# The KLF7/PFKL/ACADL axis modulates cardiac metabolic remodelling during cardiac hypertrophy in male mice

Cao Wang[1,6], Shupei Qiao[2,6], Yufang Zhao[3], Hui Tian[1], Wei Yan[4], Xiaolu Hou[5], Ruiqi Wang[1], Bosong Zhang[1], Chaofan Yang[1], Fuxing Zhu[1], Yanwen Jiao[1], Jiaming Jin[1], Yue Chen[1] & Weiming Tian [1] ✉

The main hallmark of myocardial substrate metabolism in cardiac hypertrophy or heart failure is a shift from fatty acid oxidation to greater reliance on glycolysis. However, the close correlation between glycolysis and fatty acid oxidation and underlying mechanism by which causes cardiac pathological remodelling remain unclear. We confirm that KLF7 simultaneously targets the rate-limiting enzyme of glycolysis, phosphofructokinase-1, liver, and long-chain acyl-CoA dehydrogenase, a key enzyme for fatty acid oxidation. Cardiac-specific knockout and overexpression KLF7 induce adult concentric hypertrophy and infant eccentric hypertrophy by regulating glycolysis and fatty acid oxidation fluxes in male mice, respectively. Furthermore, cardiac-specific knockdown phosphofructokinase-1, liver or overexpression long-chain acyl-CoA dehydrogenase partially rescues the cardiac hypertrophy in adult male KLF7 deficient mice. Here we show that the KLF7/PFKL/ACADL axis is a critical regulatory mechanism and may provide insight into viable therapeutic concepts aimed at the modulation of cardiac metabolic balance in hypertrophied and failing heart.

Hypertrophic cardiomyopathy (HCM) is the most common form of cardiomyopathy affecting at least 1 in 500 individuals, and a leading cause of sudden cardiac death in children and adults[1]. Cardiac hypertrophy and the progression of hypertrophy to heart failure (HF) represent major causes of morbidity and mortality, with a 5-year survival rate of 50% in adults, which is even more than that of many cancers[2–4]. Although adult and pediatric cardiomyopathies exhibit similar morphologic and clinical manifestations, their outcomes differ significantly. In HCM, 20% of infants and 4% of older children with HCM present with overt HF, and infants with HCM have a two-year mortality of 30%, whereas death is rare in older children[5]. Metabolic disorders and mitochondrial dysfunction are two of the main causes of HCM in infants and adults[6–8]. In the fetus, the heart resides in a low-oxygen environment and is therefore highly dependent on glycolysis (44%) as an ATP-generating pathway, with fatty acid oxidation (FAO) contributing only a small fraction (13%) of total myocardial ATP production. Mitochondrial FAO (70–90%) is the main source of energy in cardiomyocytes in the adult heart, and glycolysis (5%) accounts for a small proportion[9,10]. Substrate preference changes throughout the life cycles and under physiological and pathological stress. Alterations in myocardial substrate preference can be beneficial in the short term to maintain energy supply; however, they can become detrimental and result in adverse cardiac function in the long-term[11–13]. The molecular mechanisms of the close

[1]School of Life Science and Technology, Harbin Institute of Technology, 150080 Harbin, China. [2]NHC and CAMS Key Laboratory of Molecular Probe and Targeted Theranostics, Harbin Medical University, 150081 Harbin, China. [3]Space Environment Simulation Research Infrastructure, Harbin Institute of Technology, 150080 Harbin, China. [4]Department of Cardiology, The First Affiliated Hospital of Harbin Medical University, 150081 Harbin, China. [5]Department of Cardiology, The Fourth Affiliated Hospital of Harbin Medical University, 150081 Harbin, China. [6]These authors contributed equally: Cao Wang, Shupei Qiao. ✉e-mail: tianweiming@hit.edu.cn

correlation between glucose and fatty acid metabolism in the heart remain poorly understood.

Cardiac FAO enzyme expression was shown to be coordinately downregulated in murine models of ventricular pressure overload, which activates the fetal cardiac gene program, leading to increased myocardial utilization of glycolytic pathways for the production of high-energy phosphates[14–17]. The reactivation of the metabolic fetal gene program may have adverse consequences on cardiac dysfunction and eventual progression to HF[14]. Furthermore, the changes in myocardial fatty acid and glucose metabolism that occur in humans with dilated cardiomyopathy (DCM) are similar to those observed in animal models of HF. Failing human left ventricles exhibited coordinated downregulation (>40%) of medium-chain acyl-CoA dehydrogenase (MCAD) and 3-OH long-chain acyl-CoA dehydrogenase (LCHAD) at the protein and mRNA levels compared to their expression in age-matched controls. In rats with spontaneously hypertensive heart failure (SHHF), FAO enzyme mRNA levels were downregulated (>70%) compared to those of controls during both the left ventricular (LV) hypertrophy and HF stages[18], suggesting that a gene regulatory program is responsible for alterations in myocardial energy substrate utilization. Gene expression programs are controlled by thousands of transcription factors, and transcriptional regulation of metabolic function has been shown to be highly dependent on these transcription factors[19,20]. It is well established that loss-of-function mutations in certain transcription factors cause various cardiovascular deficiencies[21–23]. More importantly, a growing body of evidence demonstrates that peroxisome proliferator activated receptor (PPAR) regulate genes that encode proteins controlling metabolic homeostasis and the switch in cardiomyocyte enzymatic machinery in the heart and thereby affect pathological HF[24]. Krüppel-like factors (KLFs) are a subfamily of the zinc-finger class of transcriptional regulators. Members of this gene family have been shown to play important roles in cardiac hypertrophy and cardiac metabolism. Previous works have highlighted KLF15 deficient mice develop severe cardiac hypertrophy and heart failure under pressure overload[25–27], KLF15 is a direct and independent regulator of myocardial lipid flux and systemic metabolic homeostasis[28,29]. Similarly, KLF10 deficiency results in spontaneous pathological cardiac hypertrophy in mice at the age of 16 months[29,30]. KLF5 was shown to be a mediator of cardiac hypertrophy and fibrosis[31]. A group of studies have shown that KLF4 reactivated fetal cardiac genes during the development of cardiac hypertrophy in vivo[32–34], and KLF4 as central for transcriptional control of metabolic function and mitochondrial life cycle in the heart[35]. KLF7, a member of Krüppel-like transcription factor family[35], is a key factor in the development of the nervous system and fat formation, and is also involved in muscle regeneration and hematopoietic formation[35]. A growing body of evidence has suggested that KLF7 is associated with human obesity, type II diabetes and cardiovascular diseases, and KLF7 is one of five specific core transcription factors that regulate coronary artery disease associated pathways[36–38]. However, its role in the regulation of cardiomyopathy or cardiac metabolism has not yet been addressed[39,40]. Our work shows that KLF7 can directly target the key enzymes of cardiac glucose and fatty acid metabolism independent of the regulatory function of PPARs.

In this study, we reveal the critical transcriptional role of KLF7 in pathological cardiac hypertrophy and cardiac metabolism. We identified the rate-limiting enzyme for glycolysis, PFKL, and a key enzyme for FAO, ACADL, as direct targets of KLF7 in cardiomyocytes. KLF7 can simultaneously activate and repress the expression of ACADL and PFKL respectively to mediate the pathological progression of cardiac hypertrophy. Furthermore, both overexpression of PFKL and knockdown of ACADL led to cardiomyocyte hypertrophy, suggesting that the expression of key enzymes in glycolysis and FAO can directly induce cardiomyocyte hypertrophy, to further verify the above cellular results, we generated cardiac-specific KLF7 loss- and gain-of-function mice to explore the role and mechanism of KLF7 in pathological cardiac remodeling at different life cycles. Cardiac-specific loss of KLF7 mice function spontaneously developed severe concentric hypertrophy in adulthood, resulting in HF. KLF7 gain-of-function mice developed severe eccentric hypertrophy and cardiac dysfunction at the infant stage. Together, these data demonstrate that the KLF7/PFKL/ACADL metabolic axis is a critical regulatory mechanism for the hypertrophied and failing heart.

## Results

### Identification of PFKL and ACADL as direct targets of KLF7

To clarify whether KLF7 is involved in the process of cardiac hypertrophy, we used AngII to treat human cardiomyocyte cell line AC16 to induce cardiomyocyte hypertrophy model, and the results showed that the expression of Klf7 decreased in hypertrophic cardiomyocytes (Fig. 1a, b). In addition, we sought to determine the functional involvement of KLF7 in stress-induced cardiac hypertrophy in vivo. Thus, we subjected adult wild-type (WT) mice to transverse aortic constriction (TAC) surgery to induce a cardiac hypertrophy animal model, which caused aortic pressure to significantly increase (Supplementary Fig. 2a). Echocardiography, western blot and quantitative reverse-transcription polymerase chain reaction (qRT-PCR) also revealed that TAC surgery triggered pressure overload induced cardiac hypertrophy in the animal models ((Fig. 1h, i, Supplementary Fig. 2b–e). The transcript and protein levels of KLF7 in the myocardium were significantly decreased in comparison with those in the hearts of sham-operated mice (Fig. 1j, Supplementary Fig. 2f). In parallel, we assessed their expression in response to AngII stimulation in vitro (Fig. 1m, n, Supplementary Fig. 2g).

Then, we further identified target genes that are directly regulated by KLF7 and performed chromatin immunoprecipitation sequencing (ChIP-seq) to test whether these candidates were directly bound by KLF7 in the N2A cell line. HA-tagged Klf7 was overexpressed in cells and pulled down by HA-tag antibody. Notably, Gene Ontology (GO) functional analysis demonstrated enrichment in multiple processes, including the coenzyme metabolic process, phosphofructokinase (PFK) activity and acyl-CoA dehydrogenase (CAD) activity (Fig. 1d). Functional annotation of these enriched genes involved in metabolism revealed that glucose and fatty acid metabolic pathways, such as galactose metabolism, glycolysis and fatty acid metabolism, were affected by KLF7 (Fig. 1e). Therefore, we hypothesized that KLF7 is involved in cardiac metabolism. Recently, glycolytic flux was suggested to occur via activation of the rate-limiting enzyme PFK-1[41]. In addition, a decrease in the key enzyme ACADL was observed in HF[18], and mice with ACADL deficiency developed HCM[42]. Above all, we selected PFKL and ACADL as preliminarily targets by which KLF7 transcriptionally regulates imbalance between cardiac glucose and fatty acid metabolism. Then, we performed chromatin immunoprecipitation-polymerase chain reaction (ChIP-PCR) analysis to determine the level of PFKL and ACADL enrichment in the complex pulled down with HA-tag antibody in comparison with that pulled down with anti-IgG controls (Fig. 1g). The roles of PFKL and ACADL as KLF7 targets were further validated using luciferase assays. Transfection of a plasmid containing the luciferase sequence followed by the Pfkl and Acadl promoter regions, together with a Klf7 mimic, caused normalized luciferase activity to decrease and increase, respectively (Fig. 1f, Supplementary Fig. 1a, b), confirming that Klf7 can simultaneously positively and negatively regulate Acadl and Pfkl, respectively. Moreover, hypertrophic stimulation led to the increased expression of PFKL and decreased expression of ACADL in vivo and in vitro (Fig. 1k, l, o, p, Supplementary Fig. 2f, h).This finding was also verified quantitative analysis the Pfkl and Acadl mRNA levels in AC16 cell line under AngII stimulation (Fig. 1c). These results suggest that cardiac hypertrophic stimuli could downregulate KLF7 and ACADL expression and upregulate PFKL expression in cardiomyocytes.

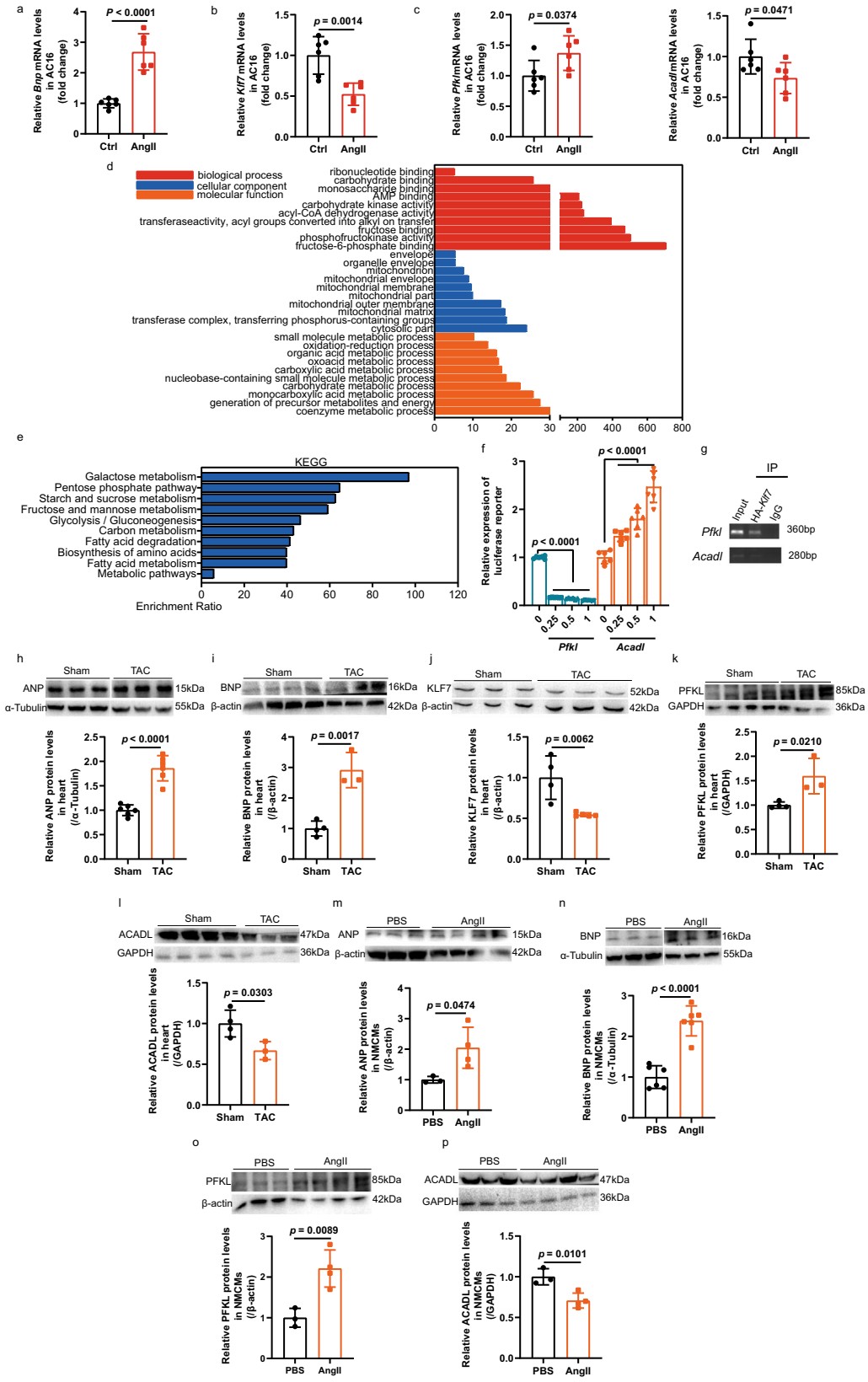

## The KLF7/PFKL/ACADL axis is involved in the pathological progression of cardiomyocyte hypertrophy

To further investigate the role of KLF7 in cellular hypertrophy, new-born mouse cardiac myocytes (NMCMs) were infected with lentivirus expressing control scrambled short hairpin RNA (shRNA) or sh-*Klf7*, followed by qRT-PCR and western blot analysis. Infection of NMCMs

with lentivirus containing sh-*Klf7* increased the mRNA and protein expression of hypertrophy-related genes with and without AngII sti-mulation (Fig. 2a, Supplementary Fig. 3b). Similarly, overexpression of PFKL by the infection of NMCMs with lentivirus expressing *Pfkl* (lenti-*Pfkl*) resulted in a hypertrophic growth phenotype (Fig. 2b, c, Sup-plementary Fig. 3i). We also demonstrated that knocking down *Acadl*

**Fig. 1 | PFKL and ACADL function as direct targets of KLF7 in the cardiac metabolism pathway. a–c** Quantitative analysis of the mRNA levels of *Bnp*, *Klf7*, *Pfkl* and *Acadl* in human cardiomyocyte cell line AC16 treated with AngII for 24 h (*n* = 6 biologically independent samples). **d, e** GO functional analysis demonstrated enrichment in phosphofructokinase activity and acyl-CoA dehydrogenase activity. KEGG analysis demonstrated enrichment in glycolysis and fatty acid metabolism. **f** *Klf7* positively regulated *Acadl* and negatively regulated *Pfkl* expression in a concentration-dependent manner, as quantified by luciferase reporter assay (*n* = 6 biologically independent experiments). **g** ChIP-PCR performed upon negative control lentivirus (lenti-NC) transfection or lentivirus-mediated HA-*Klf7* expression in NMCMs confirmed that *Klf7* binds the promoter regions of the *Pfkl* and *Acadl* genes (*n* = 2 biologically independent experiments, each independent experiment was repeated twice). **h–l** Cardiac hypertrophic growth induced by TAC downregulated the expression of KLF7, accompanied by upregulated PFKL expression and downregulated ACADL expression at the protein level (**h**, *n* = 6 biologically independent samples; **i, k, l**, sham, *n* = 4 biologically independent samples, TAC, *n* = 3 biologically independent samples; **j**, sham, *n* = 4 biologically independent samples, TAC, *n* = 5 biologically independent samples). **m–p** The protein level of PFKL was upregulated, while that of ACADL was downregulated, in NMCMs 48 h after AngII treatment (**m, o, p**, PBS, *n* = 3 biologically independent samples, AngII, *n* = 4 biologically independent samples; **n**, *n* = 6 biologically independent samples). Two-tailed unpaired Student's *t* test in **a–c** and **h–p**. One-way ANOVA with Tukey's multiple comparison test in (**f**). Data are depicted as the mean values ± SEM. KEGG Kyoto Encyclopedia of Genes and Genomes, GO Gene Ontology, NMCMs neonatal mouse cardiomyocytes, AngII angiotensin II, IP immuno-coprecipitation, TAC transverse aortic constriction. Source data are provided as a Source Data file.

triggered fetal gene expression, as indicated by the upregulation of *Anp*, *Bnp* and *Myh7* (Fig. 2d, Supplementary Fig. 3l). These results implied a close association among KLF7, PFKL, and ACADL expression and the hypertrophic growth of cardiomyocytes. In addition, we evaluated glycolysis and glycolytic reserve capacity by measuring extracellular acidification rate (ECAR), under conditions where NMCMs were supplied sequentially with glucose, oligomycin and 2-DG (Supplementary Fig. 7a), knocking down *Pfkl* displayed reduced glycolysis and glycolytic reserve capacity compared to controls (Supplementary Fig. 7b, c). To determine the impact of PFKL downregulation on cardiomyocytes FAO metabolism, we quantified cellular Oxygen consumption rate (OCR) in Supplementary Fig. 7i–m. The results demonstrated that knockdown *Pfkl* had a tendency to increase in FAO capacity, but there was no significant difference. Meanwhile, we carried out real-time respirometry in NMCMs using BSA as the substrate to assess ACADL mediated FAO (Supplementary Fig. 7d–h). As the data showed, we observed increased FAO consumption (Supplementary Fig. 7e), there were no such increase in basal respiration, ATP production and maximal respiration between overexpression of *Acadl* and controls (Supplementary Fig. 7f–h).

Then, we sought to determine whether the KLF7/PFKL/ACADL axis regulates cardiac hypertrophy growth. Based on above, we hypothesized that the knockdown of *Klf7* and *Pfkl* and overexpression of *Acadl* would result in opposite phenotypes. We also wondered whether the overexpression of *Acadl* or knockdown of *Pfkl* could attenuate cardiomyocyte hypertrophy due to the loss of *Klf7* function. To this end, we knocked down *Klf7* in NMCMs with or without exposure to AngII and subsequently transfected these cardiomyocytes with sh-*Pfkl* and lenti-*Acadl* (Fig. 2e). Notably, *Pfkl* knockdown, *Acadl* overexpression and both simultaneously were sufficient to attenuate the hypertrophic growth of AngII-treated NMCMs even in the knockdown *Klf7* (Fig. 2f–i, Supplementary Fig. 3a, j, k), suggesting that KLF7 directly bounds PFKL and ACADL to regulate cardiomyocyte hypertrophy. To fully verify the conclusion of the close association between KLF7, PFKL and ACADL, we observed overexpression of PFKL had no obvious effect on the expression of KLF7 and ACADL (Supplementary Fig. 3f, g). We also detected KLF7 and PFKL expression in knockdown *Acadl* treated NMCMs. Our results showed there was no significant difference between KLF7 and PFKL expression by knockdown *Acadl* (Supplementary Fig. 3h). The above data thus indicated that PFKL and ACADL function to mediate, at least in part, the effect of KLF7 in cardiomyocyte hypertrophic growth.

### Cardiac-specific knockout of *Klf7* in the adult heart disrupted the balance between glycolysis and fatty acid metabolism

Based on our above results, KLF7 can simultaneously regulate the key enzymes of glycolysis and FAO. Therefore, we hypothesize that KLF7 deficiency may play an important role in cardiac energy supply in adult due to upregulation of PFKL expression and downregulation of ACADL expression. We determined KLF7 could co-location in cardiomyocytes, cardiac fibroblasts, endothelial cells, and macrophages (Supplementary Fig. 5l). Therefore, we generated a mouse line with cardiac myocyte-specific deletion of *Klf7* (Myh6-*Klf7*⁻/⁻) by crossing Myh6-Cre mice with mice for which *Klf7* exon 2 was flanked with loxP sites (floxed mice) (Fig. 3a). To identify whether KLF7 mediates myocardial hypertrophy and the potential mechanism of KLF7, we performed high-throughput RNA-seq analysis to obtain transcriptomic profiles from the hearts of *Klf7*^KO (KO) and *Klf7*^fl/fl (WT) mice and compare them. We identified a total of 1713 differentially expressed genes (DEGs), of which 1531 genes were upregulated, and 182 genes were downregulated between cardiac tissues from KO and WT mice (Supplementary Fig. 4c). Notably, a close exploration of the enrichment of DEGs demonstrated extensive alterations in multiple metabolic pathways, such as lipid metabolism (Supplementary Fig. 4b). GO analysis of DEGs to assess enrichment in biological processes shows their involvement in metabolic processes (Supplementary Fig. 4a). Analysis of the top DEGs revealed that KLF7 knockout resulted in significant variations in the expression of genes related to cardiac hypertrophy, glucose and lipid metabolism, cardiac contractile function and the extracellular matrix (Supplementary Fig. 4d). Then, we focused on PFKL and ACADL and further assessed their expression in KO mice and littermate controls. We found that KLF7 deficiency resulted in a significant increase in PFKL expression and a decrease in ACADL expression (Fig. 3b, c). The use of metabolomics profiling by mass spectrometry revealed the increased production of intermediates of glycolysis in cardiac tissues deficient in KLF7 compared with their production in controls, but free fatty acids levels showed an increase, indicating that KLF7 deficiency decreased FAO capacity by inhibiting ACADL (Fig. 3h and Supplementary Fig. 5m). To directly determine the capacity of FAO and glycolysis, we examined the palmitate based OCR. Compared to WT mice, an obvious decrease in FAO consumption and respiratory capacity of mitochondria in cardiomyocytes isolated from KO mice (Fig. 3f, g). Cardiomyocyte-specific knockout *Klf7* increased glycolytic capacity compared to WT mice (Fig. 3d, e). Similar results were obtained in cardiomyocytes treated with knockdown *Klf7* and controls (Supplementary Fig. 6a–h). We next conducted a tracer experiment to further determine changes in glycolytic flux in vivo. We treated KO and WT mice with [U-¹³C] glucose and harvested the hearts 20 min later. Glycolytic intermediates, such as F-6-P, F-1, 6-BP and PDE, were significantly increased in KO mice (Fig. 3i, j). The abundance of Acetyl-CoA labeled [U-¹³C]-Palmitate was decreased in KO hearts when compared with WT hearts (Fig. 3k). Collectively, our results suggest that KLF7 deficiency leads to increased expression of rate-limiting enzymes in the glycolysis and decreased expression of key enzymes in FAO, thereby disturbing the balance between glucose and FAO in the heart.

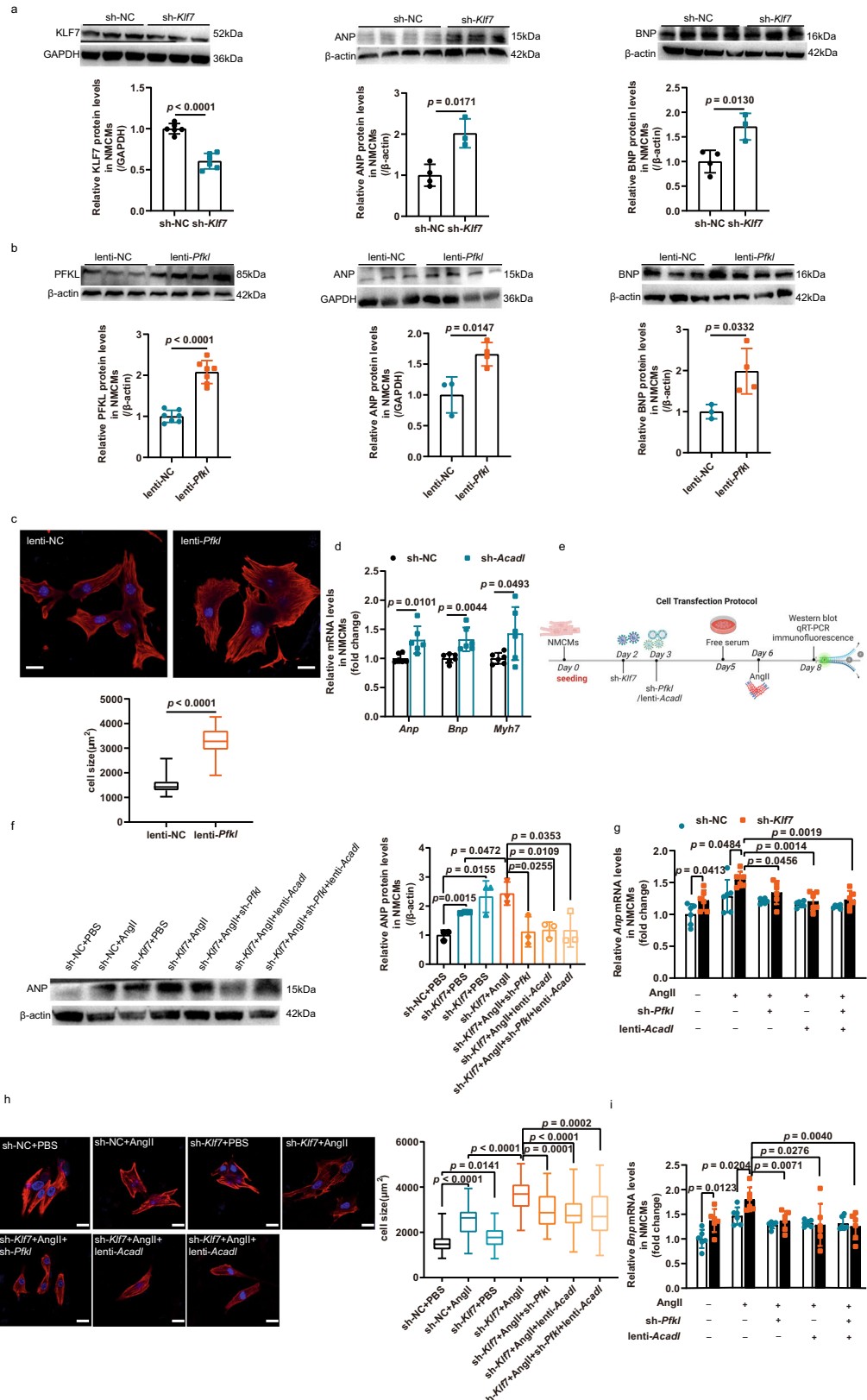

## Cardiac-specific knockout *Klf7* in adult induced concentric hypertrophy and contractile dysfunction

Male KO mice were born normally without observable differences in body weight (Supplementary Fig. 5i) relative to that of WT mice, but the heart weight to body weight ratio (HW/BW) of 6-month-old KO mice was significantly increased (Supplementary Fig. 5g). However, starting at 9-months after birth, the KO mice exhibited lethality, while WT mice did not show abnormal viability at this age (Fig. 4h). Then, we determined *Klf7* knockout efficiency in KO mice in flow cytometry sorted cardiomyocytes, cardiac *Klf7* mRNA levels were reduced by approximately 93% in the KO mice (Supplementary Fig. 5h). We also observed a significant increase in cardiac hypertrophy with age

**Fig. 2 | The PFKL/ACADL axis inhibited *Klf7* knockdown-induced cardiomyocyte hypertrophic growth. a** ANP and BNP upregulation in cardiomyocytes infected with sh-RNA targeting KLF7 (sh-*Klf7*) or sh-NC (KLF7, *n* = 6 biologically independent samples; ANP, BNP, sh-NC, *n* = 4 biologically independent samples, sh-KLF7, *n* = 3 biologically independent samples). **b** Representative western blot images and quantification of upregulated PFKL, ANP and BNP expression in NMCMs were infected by lentivirus expressing *Pfkl* (lenti-*Pfkl*) and negative control lentivirus. (PFKL, *n* = 7 biologically independent samples; ANP, BNP, lenti-NC, *n* = 3 biologically independent samples, lenti-*Pfkl*, *n* = 4 biologically independent samples). **c** Overexpression *Pfkl* mediated cardiomyocyte growth. NMCMs infected with lenti-*Pfkl* and lenti-NC. (*n* = 3; each point is the mean of *n* = 15 cells from an independent experiment); scale bar: 20 µm. **d** Quantitative analysis of the upregulation of hypertrophic markers in NMCMs infected with lentivirus to knock down ACADL (sh-*Acadl*) or sh-NC (*n* = 6 biologically independent experiments). **e** Schematic of shRNA-mediated *Klf7* and *Pfkl* knockdown and lentivirus-mediated *Acadl* overexpression in NMCMs. **f, g, i** Quantification of hypertrophic gene expression in NMCMs infected with sh-*Klf7*, sh-*Pfkl*, lenti-*Acadl* or the combination of sh-*Pfkl* and lenti-*Acadl* as indicated. *Pfkl* knockdown, *Acadl* overexpression and both simultaneously were sufficient to attenuate hypertrophic growth induced by *Klf7* knockdown (**f**, *n* = 3 biologically independent samples; **g, i**, *n* = 6 biologically independent samples). **h** *Klf7* knockdown with or without AngII stimulation, *Acadl* overexpression, *Pfkl* knockdown and simultaneous *Acadl* overexpression and *Pfkl* knockdown reduced the cross-sectional area of the NMCMs (*n* = 3; each point is the mean of *n* = 15 cells from an independent experiment); scale bar: 20 µm. **c, h** The boxplot represents the median shown as a line in the center of the box, the boundaries are the first and third quartile, and whiskers represent the minimum and maximum values in the data. Two-tailed unpaired Student's *t* test in (**a–d**), one-way ANOVA with Tukey's multiple comparison test in (**f–i**). Data are depicted as the mean values ± SEM. sh short hairpin, lenti lentivirus overexpression. Source data are provided as a Source Data file.

(Fig. 4a). Echocardiographic analysis showed that KO mice had a decreased LV posterior wall thickness at end-diastole and systole (LVPW; d/s), while both LV internal dimensions at end-diastole and systole (LVID; d/s) were significantly increased (Fig. 4c, Supplementary Fig. 5b–d). Furthermore, the KO mice exhibited impaired contractility with decreased ejection fraction (EF) and fractional shortening (FS) compared to those of the controls (Fig. 4b, Supplementary Fig. 5a). We performed Masson's trichrome staining to assess fibrosis in 9-month-old mice. The WT mice showed no sign of myocardial fibrosis, and notably, the KO mice exhibited significantly increased fibrosis in the interstitial areas of the heart (Fig. 4d). We then examined the expression of genes more directly involved in fibrosis. We observed significant increases in *Tgf-β1*, *Col I* and *Col III* (Supplementary Fig. 5k). Gross morphological analysis showed that the hearts of the KO mice were larger than those of controls, and the KO mice were characterized by both interstitial fibrosis and a disordered myocardial arrangement (Fig. 4e).

Then, we examined the effect of KLF7 deficiency on cardiac hypertrophic marker gene expression. The upregulation of atrial natriuretic peptide (ANP), brain natriuretic peptide (BNP) and myosin heavy chain (Myh7), was observed in the cardiac tissues of KO mice (Fig. 4f, Supplementary Fig. 5f). We also determined the expression of genes involved in cardiac function. The expression levels of ryanodine receptor 2 (*Ryr2*) were preserved in KO mice. Similar alterations in expression also seem to hold true for several other genes important for cardiac contractility or function, such as phospholamban (*Pln*), ATPase sarcoplasmic reticulum Ca²⁺ transporting 2 (*Serca2a*), *Tnnt2* and *Cacna1c* (Supplementary Fig. 5j). We labeled cardiomyocytes by wheat germ agglutinin (WGA) immunostaining to determine whether hypertrophy had been induced by the loss of KLF7 function. Cardiac-specific knockout *Klf7* significantly increased cardiomyocyte size (Fig. 4g). Correlation analysis indicated that tissue oxygenation was positively correlated with EF and FS, suggesting that real-time photoacoustic imaging (PAI) and its sensitivity to blood oxygenation levels are linked with cardiac function[43]. Representative PAI and qualitative observations indicated a reduction in % sO2 in the 9-month-old mice, as shown in Supplementary Fig. 5e. We observed WT mice with very high oxygen saturation (red) in the anterior myocardium, indicating high perfusion. The total red area in the same part of the hearts was reduced in the KO mice. In addition, electrocardiography (ECG) showed that the KO mice exhibited a faster heart rate at 6-month of age, but there was no significant difference in other parameters (Supplementary Fig. 8a–c). No change in blood pressure was observed, suggesting that cardiac hypertrophy was not the consequence of systemic hypertension (Supplementary Fig. 8d, e). Collectively, these findings suggest that cardiac-specific knockout KLF7 disturbed the expression of key enzymes in the glycolysis and FAO and impaired the metabolic balance of the heart, thereby inducing concentric cardiac hypertrophy.

## Cardiac-specific overexpression of *Klf7* in the infant mice heart disrupted the balance between glycolysis and fatty acid metabolism

Overexpression of *Klf7* may lead to significant inhibition of the expression of the rate-limiting enzyme in glycolysis, which may interfere with the conversion of cardiac substrates and render the energy supply insufficient during the infant period. Thus, to assess whether KLF7 can trigger pathological cardiac hypertrophy by balancing the expression of PFKL and ACADL in vivo, we generated transgenic mice (TG) in a cardiac myocyte-specific manner (Fig. 5a). Immunoblotting showed that KLF7 expression was specifically increased in cardiac tissue upon KLF7 overexpression compared to that in the controls (Supplementary Fig. 9j). Then, we thoroughly assessed the effect of *Klf7* overexpression on PFKL and ACADL. We analyzed the transcription profiles and protein levels of WT and TG mice. Compared with their expression in WT mice, overexpression of *Klf7* significantly downregulated PFKL expression and upregulated ACADL expression in 3-week TG mice after birth, and ACADL expression was reduced at 12-week, suggesting that the ability of the TG mice to carry out glycolysis and fatty acid metabolism was reduced at 12 weeks, and the mice exhibited a severe HF phenotype at this stage (Fig. 5b, c and Supplementary Fig. 9l). In addition, a thorough analysis of metabolic enzymes in the KO and TG mice revealed that variation of *Klf7* disturbed the balance between cardiac glucose and lipid metabolism (Supplementary Fig. 9k). Metabolomics analysis showed a significant decrease in glycolytic intermediates by the suppression *Pfkl* upon *Klf7* overexpression, and metabolic profiling also showed free fatty acid levels to be decreased, indicating that overexpression of *Klf7* increased FAO capacity by activating ACADL, most likely impairing cardiac energetics in the infant mice (Fig. 5h and Supplementary Fig. 9m). Seahorse analyses further confirmed that cardiomyocytes isolated from TG mice displayed reduced glycolysis compared with WT cardiomyocytes (Fig. 5d, e). In addition, TG mice exhibited a significant rise in FAO, as reflected by the FAO consumption, although there is no significant difference in basal respiration, ATP production and maximal respiration (Fig. 5f, g). As shown in metabolic flux analysis in vivo, glycolytic intermediates, such as F-6-P, F-1, 6-BP and pyruvate, were significantly decreased in TG mice (Fig. 5i). The abundance of Acetyl-CoA labeled [U-¹³C]-Palmitate was increased in TG hearts when compared with WT hearts (Fig. 5j). More importantly, In vitro level, overexpression *Klf7* NMCMs displayed a reduced rate of glycolysis compared with controls (**i** Supplementary Fig. 6i–k). Myocardial-specific overexpression *Klf7* in cardiomyocytes promoted the capacity of FAO, this was accompanied by a rise in ATP production and maximal respiration (Supplementary Fig. 6l–p). These results in combination with the metabolomics analysis indicated overexpression *Klf7* could affect the capacity of glycolysis FAO in the infant mice heart.

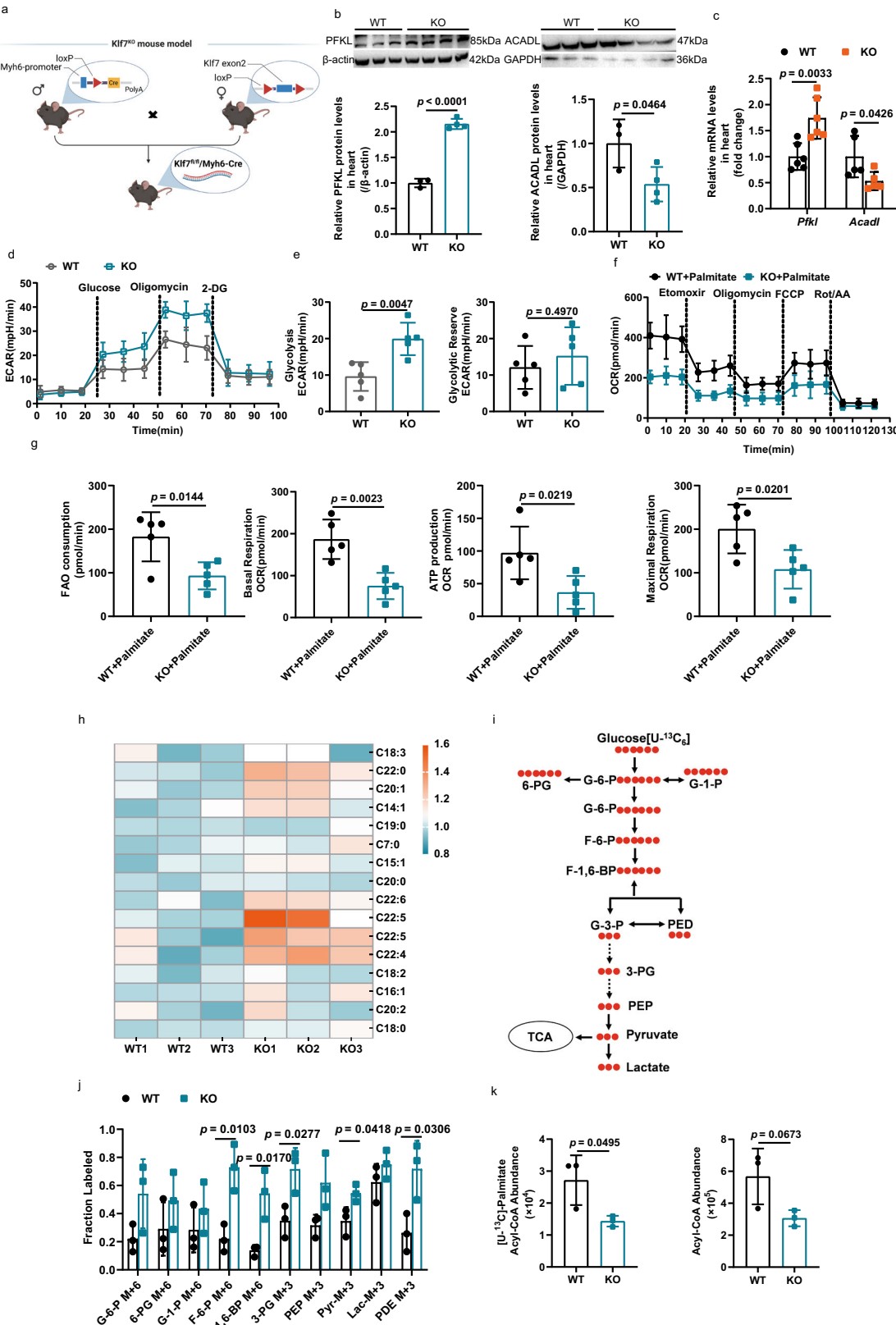

## Cardiac-specific overexpression of *Klf7* in the infant mice contributed to eccentric hypertrophy and led to HF

To determine whether cardiac metabolic imbalance impairs contractile cardiac function, WT and TG mice at different ages were subjected to echocardiogram analysis to assess cardiac function. At 3 weeks, the dysfunction progressed, with signs of HF, as shown by the significant declines in EF and FS (Fig. 6a, b, Supplementary Fig. 9a). More importantly, the LVID was increased, and the LVPW was decreased during both diastole and systole (Fig. 6c, Supplementary Fig. 9b–d). Furthermore, the TG mice exhibited a decreased interventricular septum thickness at end-diastole and systole (IVS; d/s) (Supplementary Fig. 9g, h) and tended to

**Fig. 3 | Cardiac-specific knockout *Klf7* disturbed glycolysis and FAO in adults.** **a** Schematic showing construction of the cardiomyocyte-specific *Klf7*-knockout mouse model. **b**, **c** *Klf7* deficiency upregulated *Pfkl* and downregulated *Acadl* at the mRNA and protein level, as assessed by qRT-PCR (**b**, WT, *n* = 3 biologically independent samples, KO, *n* = 4 biologically independent samples; **c**, *n* = 6 biologically independent samples). **d** Extracellular acidification rate (ECAR) of isolated cardiomyocytes using the Seahorse XF24 to analyze KO mice glycolytic function (*n* = 5 biologically independent samples). **e** KO mice have increased glycolysis and glycolytic reserve activities. (*n* = 5 biologically independent samples) **f** Oxygen consumption rate (OCR) was measured by seahorse analysis with the indicated regents in cardiomyocytes isolated from WT and KO adult mice (*n* = 5 biologically independent samples). **g** FAO consumption, Basal respiration, ATP production and Maximal respiration were decreased in KO mice compared to WT mice (*n* = 5 biologically independent samples). The glycolysis and FAO activity were analyzed by the seahorse analyzer as described in the "Methods". **h** Heat map showing that medium-, long- chain and very long-chain free fatty acid levels in the myocardial tissue were increased in *Klf7*^KO mice compared to WT mice. (*n* = 3 biologically independent samples). **i** Schematic of $^{13}$C-labeled glucose metabolic flux analysis in KO and WT mouse left ventricular tissue. **j** Quantification results of $^{13}$C-glucose metabolic flux analysis. F-6-P (m + 6), F-1, 6-BP (m + 6) and PDE (m + 3) were increased by knockout KLF7. Results are shown as fractional changes. (*n* = 3 biologically independent samples). **k** The mice intraperitoneal injection stable isotope labeled palmitate was used to calculate the contribution of FAO to acetyl-CoA entering the TAC. The abundance of [U-$^{13}$C]-Palmitate labeled acetyl-CoA decreased in KO mice compared to WT mice. (*n* = 3 biologically independent samples). Two-tailed unpaired Student's *t* test in (**b**, **c**, **e**, **g**, **j**, **k**). Data are depicted as the mean values ± SEM. FAO, fatty acid oxidation; KO, knockout of *klf7* mice. Source data are provided as a Source Data file.

exhibit a larger chamber size and LV volume (LVvol) (Supplementary Fig. 9e, f).

In addition, TG mice showed substantial increases in HW/BW and heart size (Fig. 6i, Supplementary Fig. 9i). Histological analysis demonstrated that the cardiac ventricular wall and IVS were thinner in the hearts of TG mice compared with WT mice, and TG mouse hearts exhibited a disordered myocardial arrangement (Fig. 6e). Masson's trichrome staining revealed massive fibrosis in the TG mouse myocardium (Fig. 6d). Furthermore, cardiac overexpression of *Klf7* significantly increased cardiomyocyte size (Fig. 6h). Similarly, we also found increased expression of ANP, BNP, and Myh7 in the TG mice compared to the WT mice (Fig. 6f, g). Unlike the WT mice, the TG mice began to die at 3 weeks of age (Fig. 6j). Above all, the cardiac-specific overexpression of *Klf7* disturbed the expression of enzymes involved in glucose and fatty acid metabolism during the infant stage, thereby inhibiting the main source of energy during this period (glycolysis), resulting in eccentric cardiac hypertrophy.

## Cardiac-specific knockdown of *Pfkl* or overexpression of *Acadl* rescued cardiac hypertrophy and myocardial fibrosis in adult KO mice

At the animal level, we verified whether knockdown of *Pfkl* or overexpression of *Acadl* could alleviate the pathological phenotypes of cardiac hypertrophy and HF caused by cardiac-specific knockout *Klf7*. Since hearts from 6-month-old KO mice showed obvious cardiac hypertrophy, postnatal 5-month-old KO mice were tail-vein injected with knockdown NC and *Pfkl* or overexpression NC or *Acadl* for 4 weeks, after which the mice were subjected to echocardiography (Figs. 7a and 8a). The expression levels of PFKL and ACADL were reduced to 50% and increased by 200%, respectively, by injection of 3 E + 11 adeno-associated virus serotype 9 (AAV9) virus particles compared to their expression in the controls (Supplementary Fig. 10b, c, h). There was no significant difference in *Pfkl*-knockdown WT mice compared to NC-knockdown WT mice in (%) EF (Supplementary Fig. 10d). In KO mice, knockdown of *Pfkl* or overexpression of *Acadl* improved cardiac function, as shown by the LVvol and LVID during systole and diastole (Fig. 7b, c and Fig. 8b, c). Other echocardiography parameters tended to normalize, but there was no significant difference in *Pfkl*-knockdown KO mice compared to KO mice without *Pfkl* knockdown (Supplementary Fig. 10d–g). However, *Acadl* overexpression restored cardiac function in the KO mice to a greater extent (Supplementary Fig. 10i–k). Consistently, knockdown of *Pfkl* or overexpression of *Acadl* decreased the relative HW/BW in the KO mice (Figs. 7e, 8f). Knockdown *Pfkl* also decreased the heart volume of KO mice (Supplementary Fig. 10a). In addition, knockdown *Pfkl* or overexpression *Acadl* improved fibrosis and the disordered myocardial tissue arrangement in the hearts of KO mice (Figs. 7d, g, 8e, g), and suppressed knockout of *Klf7* induced enlargement of myocyte (Fig. 7f, 8h). Similarly, the increases in *Anp* and *Bnp* levels in the hearts of KO mice were significantly abrogated by *Pfkl* knockdown or *Acadl* overexpression (Fig. 7o, 8q).

To assess whether the observed knockdown *Pfkl* and overexpression *Acadl* were correlated with glycolysis and FAO function, respectively. Metabolic flux analysis was performed, the OCR and ECAR were measured in WT, KO and KO-AAV-*Pfkl* and KO-AAV-*Acadl* mice heart using Seahorse XF24 analyzer. The results showed that knockdown *Pfkl* significantly decreased glycolysis in the KO-AAV-*Pfkl* mice heart compared with KO mice (Fig. 7h, i), however, there was no significant difference between KO and KO-AAV-*Pfkl* mice (Fig. 7j–n). In addition, myocardial-specific overexpression *Acadl* (KO-AAV-*Acadl*) exhibited enhanced FAO capacity, as reflected by increased FAO consumption (Fig. 8i–m). Next, we evaluated the ECAR in the overexpression *Acadl* in KO mice. As shown in the Fig. 8n–p, overexpression *Acadl* did not impact glycolysis and glycolytic reserve capacity. Taken together, these findings suggest that the knockdown *Pfkl* or overexpression *Acadl* partially restored cardiac function in the KO mice, and that KLF7 can simultaneously regulate the expression of PFKL and ACADL to restore cardiac metabolic balance and improve cardiac function.

## Discussion

We have shown that knockout or overexpression *Klf7* can induce cardiac hypertrophy and HF by disturbing the cardiac metabolic balance and serve as a metabolic molecular switch to repress and activate genes involved in cardiac glycolysis and FAO. Evidence has previously shown that PPARs are mainly master switches in cardiac lipid metabolism; thus, PPARs are now considered to be attractive targets for the development of therapeutic strategies against metabolic syndrome[44]. KLF15 is required for the ability of PPARα to induce expression of a subset of target genes critical for cardiac lipid oxidation[45]. KLF5 co-regulates PPAR-δ, functioning as a molecular switch for FAO programs[46]. Here, we found that KLF7 can directly target the expression of key enzymes in glycolysis and fatty acid metabolism independent of the function of PPARs and thereby affect the cardiac metabolic balance.

A study showed that glycolytic flux activates PFK-1 by increasing the concentrations of its activators and glucose transport in hypertrophic hearts[17,41]. In addition, another study indicated that exercise increased glycolysis via PFK-1 phosphorylation, promoting physiological cardiac hypertrophy[47]. However, the direct correlation between PFKL expression and cardiac hypertrophy has remained unclear. Here, our findings show that PFKL was induced in vivo and in vitro in a cardiac hypertrophy model. Then, we demonstrate that chronic, persistent induction of PFKL in the heart directly drives cardiac hypertrophy by activating the glycolysis. Moreover, cardiac-specific PFKL knockdown improved the pathological phenotype of cardiac hypertrophy in KO mice. Therefore, persistently elevated PFKL expression may be the mechanism underlying pathological cardiac remodeling. Using both in vitro and in vivo models of hypertrophy, we found that the ACADL level was significantly decreased in cardiomyocytes during hypertrophic growth. This finding is consistent with prior studies

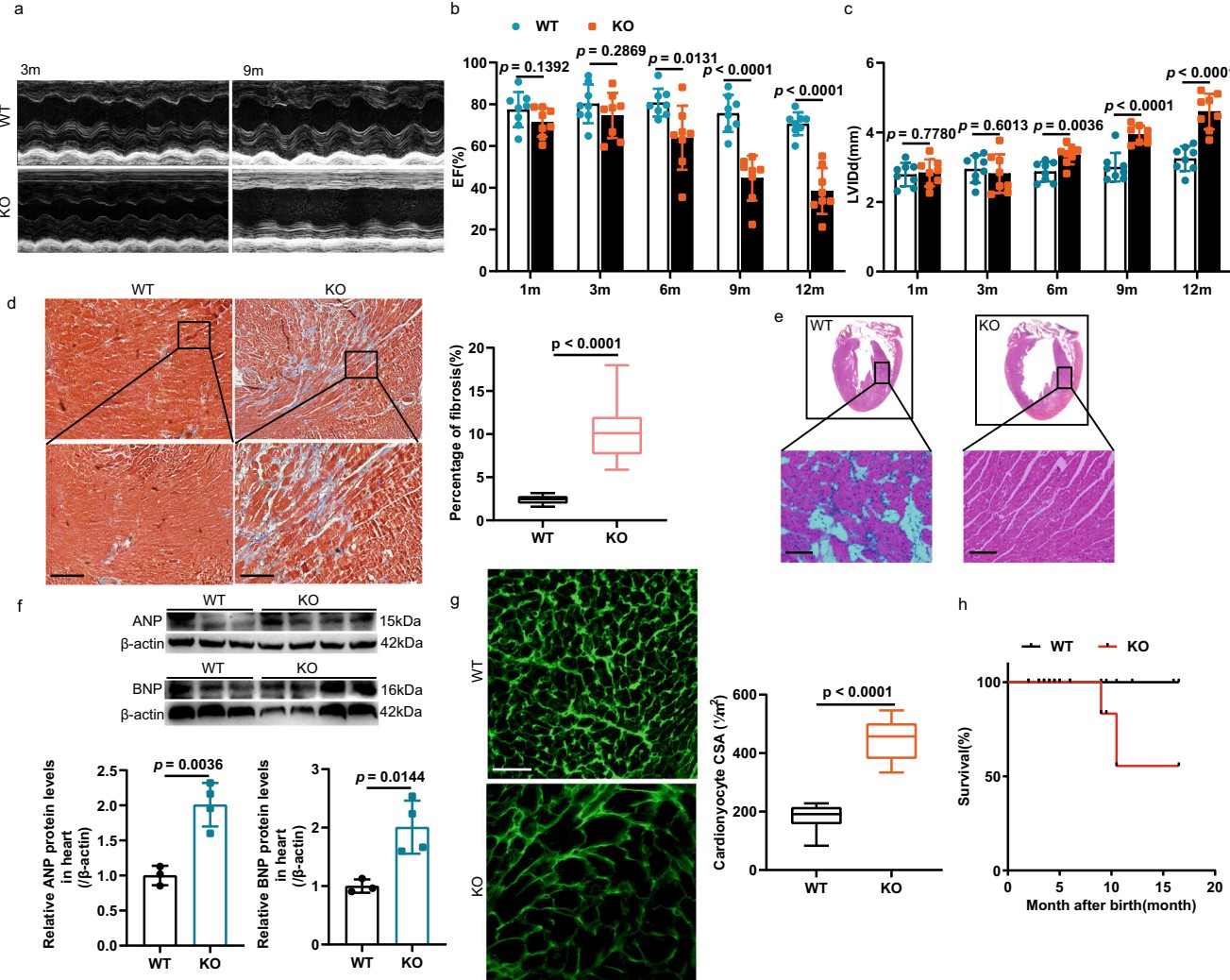

**Fig. 4 | KO mice displayed cardiac hypertrophy and HF.** KO mice and WT mice of different ages were compared. **a** Representative examples of longitudinal M-mode echocardiography in 3-month-old and 9-month-old KO and WT mouse hearts are shown. **b** Decreased ejection fraction (EF) ($n = 8$ mice/group). **c** Increased end-diastolic left ventricle inner diameter ($n = 8$ mice/group). **d** Masson's trichrome staining analysis of myocardial tissue with severe fibrosis ($n = 3$ mice/group, 8 sections/mouse); scale bar: 500 μm. **e** Analyses of gross heart morphology and H&E-stained longitudinal sections of myocardial tissue showing a disordered myocardial arrangement (n = 3 mice/group, 8 sections/mouse); scale bar: 100 μm. **f** Western blot analysis showing the increased expression of cardiac hypertrophy markers in cardiac tissue (WT, $n = 3$ biologically independent samples; KO, $n = 4$ biologically independent samples). **g** Immunofluorescence staining for wheat germ agglutinin (WGA) was used to analyze the increased cross-sectional area of cardiomyocytes, ($n = 3$ mice/group, 9 sections/mouse); scale bar: 100 μm. **h** The survival rate of KO mice and WT mice of different ages were decreased ($n = 13$ mice/group). **d**, **g** The boxplot represents the median shown as a line in the center of the box, the boundaries are the first and third quartile, and whiskers represent the minimum and maximum values in the data. Two-way ANOVA with Tukey's multiple comparison test in (**b**, **c**). Two-tailed unpaired Student's $t$ test in (**d**, **f**, and **g**). Data are depicted as the mean values ± SEM. H&E hematoxylin and eosin, CSA cell cross-sectional area, HW/BW heart weight to body weight ratio. Source data are provided as a Source Data file.

showing that FAO is low and associated with decreased expression of regulatory enzymes in hypertrophy and in the fetal heart[14,18]. Supporting our findings, an early study showed a decrease in ACADL in HF. Mice deficient in ACADL were also found to develop metabolic cardiomyopathy, which manifested as cardiac hypertrophy[36]. Our findings show that KLF7/PFKL/ACADL signaling has been shown to be an essential process for cardiac FAO and glycolysis, leading to cardiac hypertrophy and HF. However, previous work has focused on the relative oxidation of only fatty acids or glucose in the heart[7,48]. We aimed to maintain the balance of overall cardiac ATP generation regulated by enzymes involved in glycolysis and FAO. The regulatory pattern that we have revealed will have a beneficial effect on understanding the pathological process of myocardial hypertrophy. Although we focused on PFKL and ACADL in the present study, it is very likely that the expression of other genes is also controlled by

transcriptional programs involving KLF7 to induce cardiac hypertrophy.

Although changes in cardiac metabolism during hypertrophy and chronic HF are well documented, these changes are often viewed as a compensatory or merely coincidental phenomenon rather than a cause of cardiomyopathy[49–52], thus, the question of a casual role of alterations in myocardial metabolism has been controversial for decades. Several lines of evidence suggest that changes in myocardial fuel utilization are an early event that occurs with the hypertrophic growth response and subsequent remodeling[53–55]. Moreover, many studies have found that inhibition of glycolysis and/or stimulation of FAO could alleviate the pathological remodeling of cardiac hypertrophy or HF [56]. Cardiac-specific deletion of ACC2 which results in high cardiac FAO rates, are protected against the development of cardiac hypertrophy[57]. Astragaloside IV switched glycolysis to FAO and

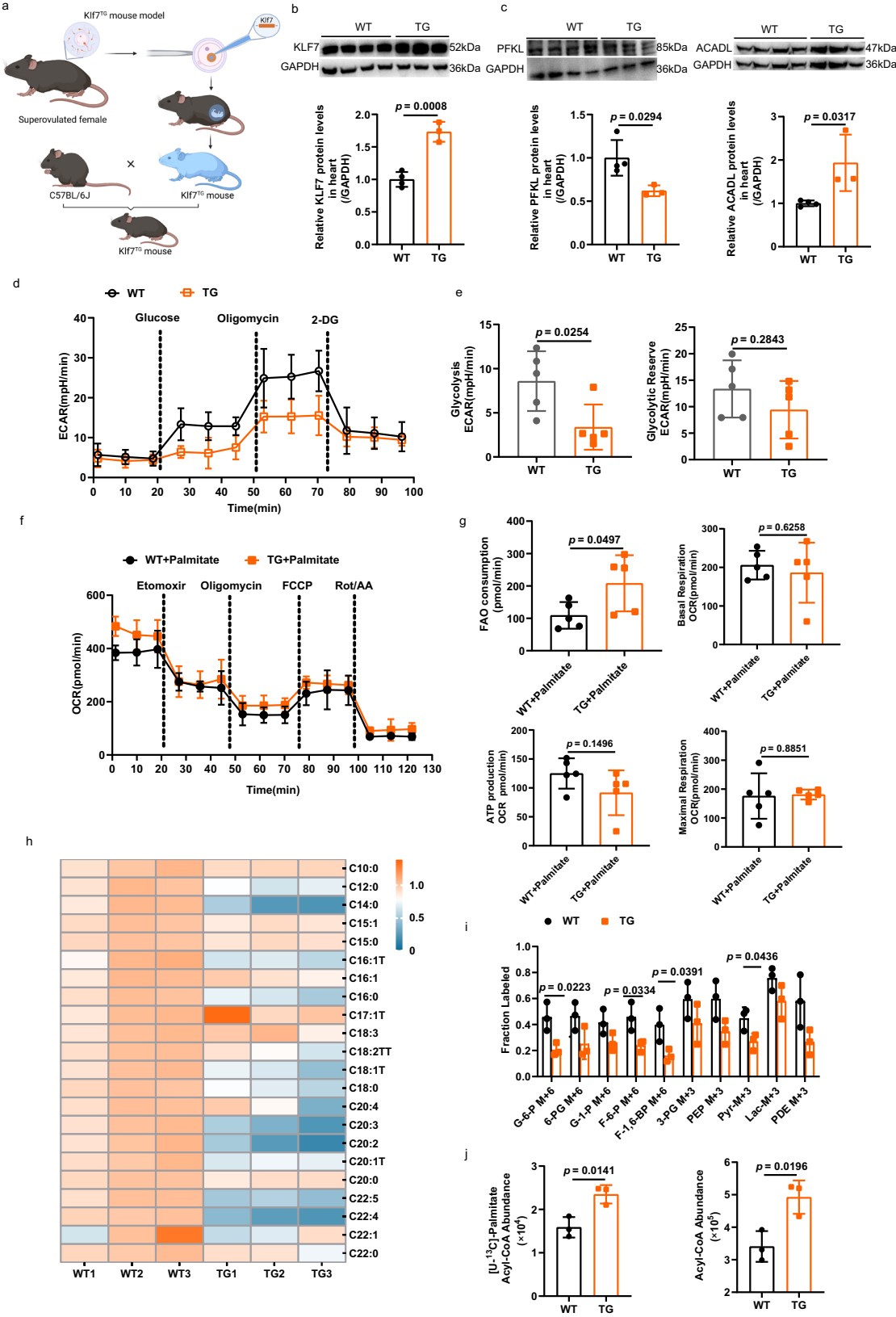

improved mitochondrial function, which may present a cardio-protective treatment that inhibits the progress of HF[58]. Empaglifloxin reduces glycolysis and rebalances coupling between glycolysis and oxidative phosphorylation to attenuate adverse cardiac remodeling and progression of HF[59]. Serpina3c inhibits glycolysis and provides a potential intervention target for the treatment of HF[60]. Inhibition of

AMP-activated kinase in vivo prevented enhanced glycolytic flow and rescued cardiac hypertrophy[61]. Here, we provide direct evidence that the disruption of glycolysis and FAO was the cause of contractile dysfunction in KO and TG mice. We confirmed that KO and TG mice disturbed the expression of key enzymes in glycolysis and FAO at the adult and infant stages, respectively, which in turn affected cardiac

**Fig. 5 | Cardiac-specific overexpression of *Klf7* disturbed the balance between glycolysis and FAO in the infant mice. a** Schematic showing construction of the cardiomyocyte-specific *Klf7*-overexpression mouse model. **b, c** The protein level of the KLF7 target gene PFKL was downregulated, and the ACADL protein level was upregulated in myocardial tissues of TG mice of 3 weeks compared to their expression in corresponding WT mice (WT, $n = 4$ biologically independent samples, TG, $n = 3$ biologically independent samples). **d** ECAR of isolated cardiomyocytes using the Seahorse XF24 to analyze 3 weeks TG mice glycolytic function ($n = 5$ biologically independent samples). **e** TG mice have decreased glycolysis and glycolytic reserve activities. ($n = 5$ biologically independent samples). **f** OCR was measured by seahorse analysis in cardiomyocytes isolated from WT and TG 3 weeks mice ($n = 5$ biologically independent samples). **g** FAO consumption, Basal respiration, ATP production and Maximal respiration in WT and TG mice ($n = 5$ biologically independent samples). **h** Heat map showing that medium-, long-chain and very long-chain free fatty acid levels in the myocardial tissues were decreased TG mice compared to WT mice ($n = 3$ biologically independent samples). **i** Quantification results of $^{13}$C-glucose metabolic flux analysis. F-6-P (m + 6), F-1, 6-BP (m + 6) and Pyruvate (m + 3) were decreased by overexpression *Klf7*. Results are shown as fractional changes. ($n = 3$ biologically independent samples). **j** The abundance of [U-$^{13}$C]-Palmitate labeled acetyl-CoA increased in TG mice compared to WT mice. ($n = 3$ biologically independent samples). Two-tailed unpaired Student's $t$ test in (**b**, **c**, **e**, **g**, **i** and **j**). Data are depicted as the mean values ± SEM. TG, overexpression of *klf7* mice. Source data are provided as a Source Data file.

glycolysis and FAO, leading to cardiac contractile dysfunction. Therefore, targeting the metabolic balance may provide a novel therapeutic strategy for the cardiac hypertrophy and HF. Furthermore, in our study, we demonstrated at both in vitro and in vivo that knockdown *Pfkl* and overexpression *Acadl* partially alleviated the phenotype of cardiac hypertrophic growth induced by knockdown *Klf7* by inhibiting glycolysis and enhancing FAO, respectively. Understanding the molecular basis of metabolic homeostasis will pave the way for the development of therapies for HF.

The following animal models have been extensively used to study pathological cardiac processes. Adenine nucleotide transporter (ANT) deficient mice exhibited the phenotypes of mitochondrial dysfunction and cardiac hypertrophy[62]. Mice deficient in mitochondrial transcription factor A (Tfam) were found to die in the neonatal period[63], and very-long-chain acetyl-CoA dehydrogenase (VLCAD) deficient mice showed lipid aggregation and myocardial fibrosis[42]. However, these currently available animal models test the effects of only unbalanced glucose metabolism or FAO. The animal model of infant or juvenile cardiac hypertrophy is still lacking. A previous study confirmed that juvenile visceral steatosis in mice, a genetic model of systemic carnitine deficiency, resulted in disordered mitochondrial β-oxidation and led to the development of marked cardiac hypertrophy[64,65]. Here, under basal conditions, KO and TG mice were distinguishable from WT mice in terms of heart morphology, cardiac function and the expression profiles of genes involved in cardiac metabolism. Therefore, both KO and TG mice could be used as models of spontaneous cardiac hypertrophy and even HF to study the pathological progression of these related metabolic diseases from the infant to adult stages.

In conclusion, our findings show that KLF7 is a unique regulator of metabolism from the infant to adult stages and serves as a molecular switch that simultaneously represses and activates genes involved in glycolysis and FAO, KLF7 may be an attractive therapeutic target for metabolic syndrome at different life cycles.

## Methods
### Animals
All animal experiments were approved and performed in accordance with guidelines set forth by the Harbin Institute of Technology Committee on Animal Resources (protocol number: IACUC-2020035). Female and Male C56BL/6 J mice (6–8 weeks old), Female KO mice (6–8 weeks old), male αMHC-Cre mice (6–8 weeks old) and male TG mice (6-8 weeks old) were purchased from Cyagen Co., Ltd were applied for fertile. To generate cardiomyocyte-specific knockout *Klf7* mice, mice harboring a floxed *Klf7* allele were crossed with αMHC-Cre mice. Cardiomyocyte-specific *Klf7*-overexpression mice were generated by purifying the αMHC-*Klf7* vector, inserting it into linearized DNA and microinjecting the DNA into the fertilized ovum. KO, TG and corresponding control floxed or αMHC-Cre mice were maintained on a C57BL/6J genetic background. The KO and WT mice (1-12 months) and TG and WT mice (3–12 weeks) were usually applied for the study of experiment. We used male mice in all experiments to avoid the effects of changes in female sex hormone levels during aging on myocardial remodeling and metabolism. Therefore, our findings may not apply to female mice. The diet for the mice was purchased from Shanghai R&S Biotechnology Co., Ltd (M2119). All the mice were kept in an environment that was free of pathogens (at 22–25 °C, relative humidity of 45–60%, a 12 h light/dark cycle). The primer sequences for mouse genotyping are as follows: *Klf7*, forward 5′-GAGCAGTCTATTTGCATCTTGCTT-3′ and reverse 5′-AAGGCCACATGGAATGACTTTAAC-3′; Cre, forward 5′-TCTATTGCACACAGCAATCCA-3′ and reverse 5′-CCAGCATTGTGAGAACAAGG-3′; TG, forward 5′-AGAGAAGCAGGCACTTTACATGG-3′ and reverse 5′-GGGGTTCTGTCTGGAGGTAGCGT-3′.

### Echocardiography
Transthoracic echocardiography was performed on conscious, gently restrained mice using the VisualSonic Vevo 3100 system. The data were processed by Vevo LAB (version 3.2.0) software. Male mice were anaesthetized with 5% isoflurane and then restrained on the platform under 0.25-0.5% isoflurane. The left ventricle was assessed in both parasternal long-axis views. LV wall thickness, the LV diameter, the LV volume, the EF and FS were measured from M-mode at 1 kHz at the papillary muscle level.

### TAC surgery
Male mice (8-10 weeks old) of the respective genotypes with normal FS were subjected to pressure overload by TAC and constricted against a 27-gauge needle. The sham control animals underwent exactly the same procedure except for aortic constriction. At defined time points, animals were euthanized, and whole hearts or left ventricles were collected for histological, molecular, and biochemical analyses.

### Neonatal cardiomyocyte isolation and lentivirus infection
Primary NMCMs were isolated and cultured in Dulbecco's modified Eagle's medium (DMEM) containing 10% fetal bovine serum (FBS) and antibiotics. For lentivirus-mediated gene knockdown and overexpression, cells were incubated for 6 h with lentivirus at a multiplicity of infection of 50 plaque-forming units per cell. NMCMs were infected with lentivirus for 48 h, and the infection efficiency was then analyzed by measuring green fluorescent protein (GFP) fluorescence. The shRNA plasmid was purchased from Shanghai Genechem Co., Ltd (sh-*Pfkl*, catalog no. GIEE0269091; sh-*Acadl*, catalog no. GIEE0269090; sh-*Klf7*, catalog no. GIEE0269089).

### Western blotting
Whole-cell lysates from mouse left ventricles and NMCMs were prepared in RIPA buffer (Millipore, catalog no. 20-188) containing protease inhibitor cocktail (Thermo Fisher Scientific, catalog no. 78425). Samples containing an equal amount of protein (10-20 μg) were separated on a 12% SDS-PAGE gel (Thermo Fisher Scientific, catalog no. XP00122BOX), transferred to a nitrocellulose membrane, and immunoblotted, after which specific protein bands were detected and quantified using an Odyssey scanner (version 3) and Image J (version 2), separately. Proteins were detected with primary antibodies including

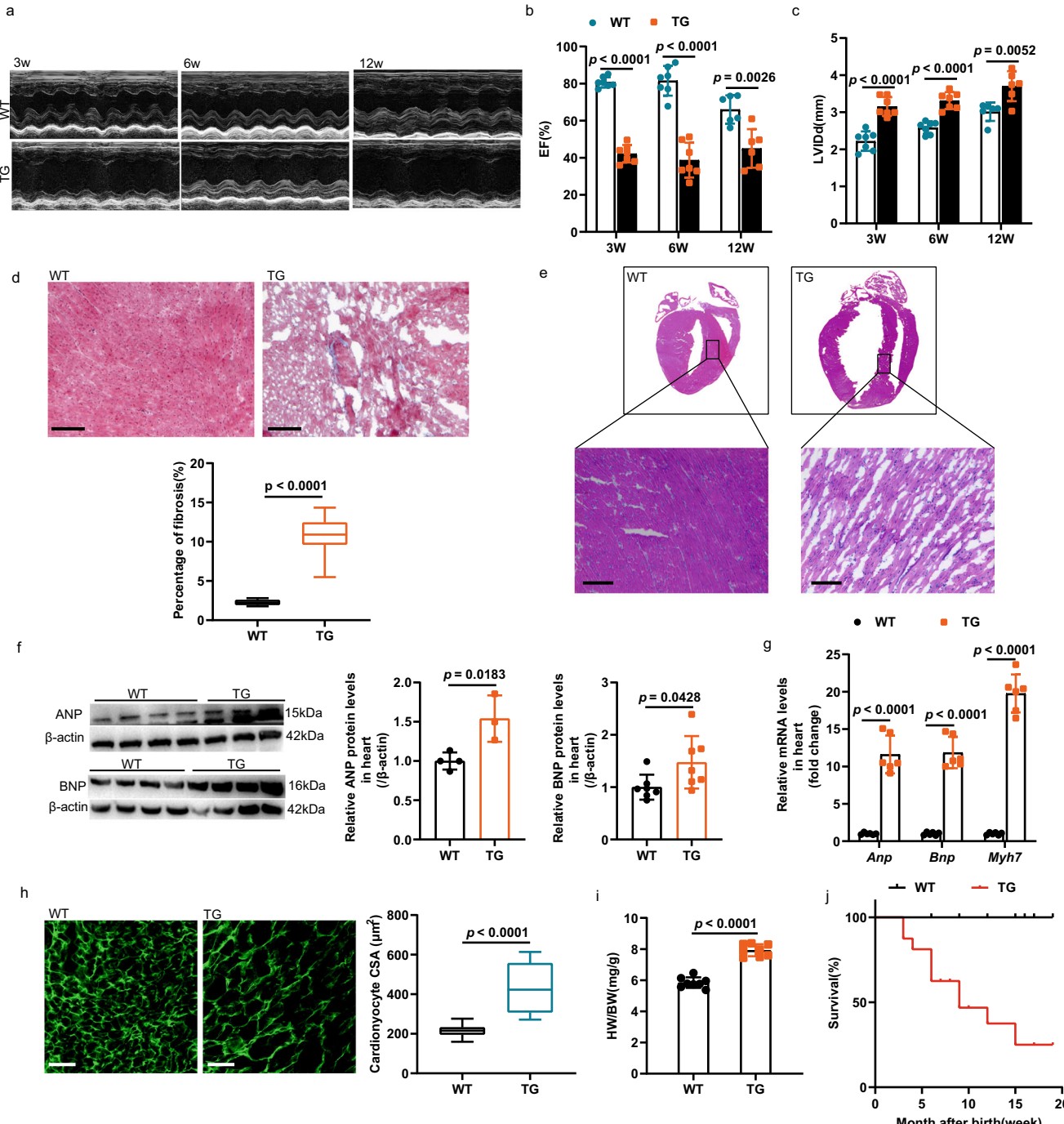

**Fig. 6 | Mice with cardiomyocyte-specific overexpression of *Klf7* displayed cardiac hypertrophy and myocardial fibrosis. a** Long axis transthoracic M- mode echocardiographic traces from WT and TG mice at 3, 6, and 12 weeks of age. Comparison of TG mice with WT mice revealed changes in echocardiographic parameters: **b** a decrease in EF (*n* = 7 mice/group) and **c** an increase in LVIDd (*n* = 7 mice/group). **d** Masson's trichrome staining analysis of myocardial tissue revealed severe fibrosis, (*n* = 3 mice/group, 8 sections/mouse); scale bar: 500 μm. **e** Disordered myocardial tissue arrangement (*n* = 3 mice/group, 8 sections/mouse), scale bar: 100 μm. **f, g** Comparison of TG mice with WT mice revealed the expression of fetal genes to be upregulated (**f**, ANP, WT, *n* = 4 biologically independent samples, TG, *n* = 3 biologically independent samples; BNP, *n* = 7

biologically independent samples; **g**, *n* = 6 biologically independent samples). **h** WGA staining analysis revealed the increased cross-sectional area of cardiomyocytes, (*n* = 3 mice/group, 8 sections/mouse); scale bar: 100 μm. **i** The HW/BW was increased in 6-week-old TG mice compared to WT mice (*n* = 8 mice/group). **j** The survival rate of the TG mice was lower (n = 12 mice/group). **d, h** The boxplot represents the median shown as a line in the center of the box, the boundaries are the first and third quartile, and whiskers represent the minimum and maximum values in the data. Two-way ANOVA with Tukey's multiple comparison test in (**b, c**). Two-tailed unpaired Student's *t* test in (**d**) and (**f–i**). Data are depicted as the mean values ± SEM. CSA cell cross-sectional area, HW/BW heart weight to body weight ratio. Source data are provided as a Source Data file.

Anti-ANP Rabbit pAb (1:1000; Abcam, catalog no. ab180649), Anti-BNP Rabbit pAb (1:500, ABclonal, catalog no.A2179), Anti-ACADL Rabbit pAb (1:1000, Abcam, catalog no. ab129711), Anti-PFKL Rabbit pAb (1:1000, GeneTex, catalog no. GTX105697), Anti-KLF7 mouse mAb

(1:1000, Abnova, catalog no. H00008609-M01, clone no. 3E8-B8), Anti-α-Tubulin mouse mAb (1:2000, ABclonal, catalog no. AC012, clone no. AMC0479), Anti-β-actin Rabbit pAb (1:1000, ABclonal, catalog no. AC006) and Anti-GAPDH Rabbit pAb (1:1000, ABclonal, catalog no.

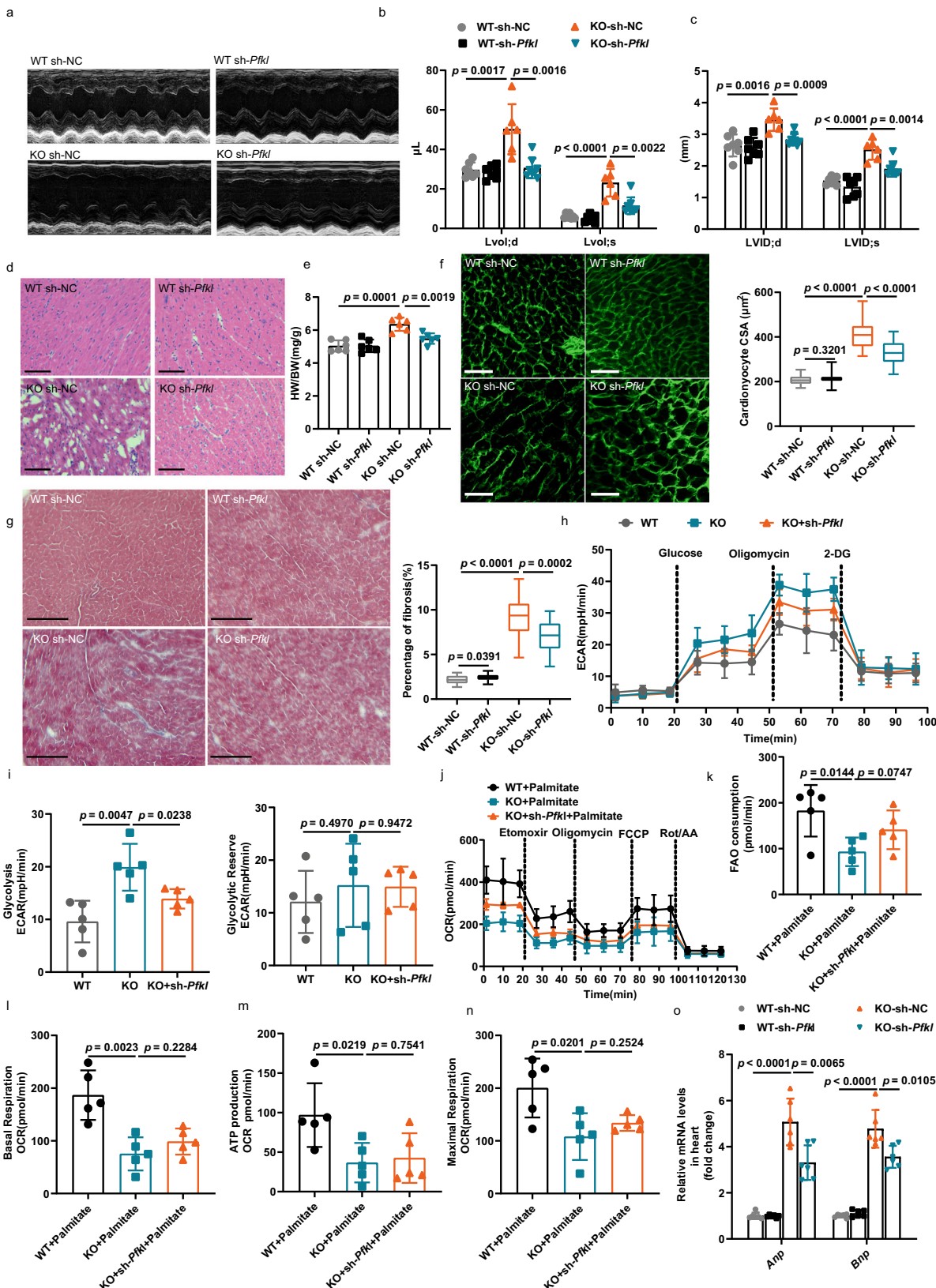

AC001). The uncropped blots for the main and supplementary figures are provided in the Source Data file.

## Histology and immunohistochemistry

Zeiss LSM980 confocal microscope (ZEISS) and Stereo Discovery.V8 were used to acquire fluorescence signals and tissue morphology. The images were processed by ZEN software (blue edition). Histology and immunohistochemical (IHC) analyses were performed with formaldehyde-fixed hearts that were routinely processed to generate frozen sections. Briefly, hearts were fixed in 4% paraformaldehyde overnight at 4 °C and transferred to ×PBS, followed by optimal cutting temperature compound (OTC) embedding. Haematoxylin and Eosin

**Fig. 7 | Cardiac-specific knockdown of *Pfkl* protected *Klf7*-deficient hearts from hypertrophic cardiomyopathy. a** Representative echocardiogram of 6-month-old mice treated with AAV-NC or sh-*Pfkl* for 4 weeks. **b, c** Cardiac-specific knockdown of *Pfkl* (sh-*Pfkl*) significantly abrogated the increases in LVvol and LVID during systole and diastole induced by cardiac-specific *Klf7* deficiency in the mice (*n* = 6 mice/group). **d** sh-*Pfkl* improved the disordered myocardial tissue arrangement in *Klf7*$^{KO}$ mice (*n* = 3 mice/group, 7 sections/mouse); scale bar: 100 μm. **e** sh-*Pfkl* decreased the HW/BW in *Klf7*$^{KO}$ mice (*n* = 6 mice/group). **f** sh-*Pfkl* reduced the enlargement of cardiomyocytes induced by knockout *Klf7*, as shown by WGA staining, (*n* = 3 mice/group, 9 sections/mouse); scale bar: 100 μm. **g** Masson's trichrome staining analysis revealed a reduction in myocardial fibrosis in the *Klf7*$^{KO}$ mice, (*n* = 3 mice/group, 8 sections/mouse); scale bar: 500 μm. **h–i** ECAR in cardiomyocytes isolated from mice hearts were measured under basal, glucose (10 mM), oligomycin (1 μM) and 2-DG (50 mM) stimulated conditions at the indicated time points.

Quantification of the glycolysis and glycolytic reserve (*n* = 5 biologically independent samples). **j** OCR in cardiomyocytes isolated from mice hearts were measured under basal, Etomoxir (40 μM), oligomycin (1 μM), FCCP (1 μM), and Rot/AA (0.5 μM) conditions at the indicated time points (*n* = 5 biologically independent samples). **k–n** Quantification of the FAO consumption and mitochondrial basal respiration, ATP production and maximal respiration rate (*n* = 5 biologically independent samples). **o** qRT-PCR analysis showed that sh-*Pfkl* decreased expression of the hypertrophic genes *Anp* and *Bnp* (*n* = 6 biologically independent experiments). **f, g** The boxplot represents the median shown as a line in the center of the box, the boundaries are the first and third quartile, and whiskers represent the minimum and maximum values in the data. One-way ANOVA with Tukey's multiple comparison test in (**b**, **c**, **e–g**, **i**, and **k–o**). Data are depicted as the mean values ± SEM. CSA, cell cross-sectional area; HW/BW, heart weight to body weight ratio; sh, knockdown with adeno-associated virus. Source data are provided as a Source Data file.

staining was performed for morphological analysis. WGA staining was used for cross-sectional area (CSA) measurements for at least 30 cells per section and three independent heart sections per group. Masson's trichrome staining was used to measure fibrosis. IHC analyses of Anti-cardiac troponin T mouse mAb (1:200, Abcam, catalog no. ab8295, clone no. 1C11), Anti-alpha smooth muscle Actin rabbit mAb (1:200, Abcam, catalog no. ab124964, clone no. EPR5368), Anti-CD31 rabbit mAb (1:200, Abcam, catalog no. ab222783, clone no. EPR17260-263), Anti-CD68 rat mAb (1:200, Abcam, catalog no. ab53444, clone no. FA-11) and Anti-Vimentin rabbit mAb (1:200, Abcam, catalog no. ab92547, clone no. EPR3776) were performed to detect the cardiomyocyte CSA and cell co-localization, and DAPI was used to stain nuclei.

### Cell lines

Human cardiomyocyte AC16 cells (catalog no. BFN60808678) and human embryonic kidney HEK293T cells (catalog no. BFN60810479) were acquired from the Cell Bank of the Shanghai Academy of Chinese Sciences and were cultured in Dulbecco's modified Eagle's medium (Gibco, catalog no.10566016) that contained 10% fetal bovine serum (Gibco, catalog no.10099141) and 1% penicillin-streptomycin (Invitrogen, catalog no.15140155).

### Luciferase and ChIP assay

Using pGL3, a luciferase vector harboring two TREs with or without constitutively active *Klf7* expression vectors luciferase assays was performed in HEK293T cells. Plasmids were transfected into HEK293T cells using Lipofectamine 3000 (Invitrogen). Luciferase assays using *Pfkl* or *Acadl* reporter constructs of different lengths harboring *Klf7*-response elements (FREs) were performed in HEK293T cells with or without *Klf7* expression. Luciferase activity was measured using a dual-luciferase kit (Promega) and normalized to Renilla activity.

Quantitative analyses of ChIP assay data to assess KLF7 occupancy at the *Pfkl* or *Acadl* promoter in NMCMs were conducted using the Zymo-Spin ChIP kit (Epigenetics) according to the manufacturer's instructions. Briefly, isolated NMCMs overexpressed HA-tagged *Klf7* for 48 h. After the crosslinking reaction was quenched with glycine, the tissue was washed three times with PBS and then homogenized in nuclear extraction buffer. The homogenate was centrifuged (10 min, 500 × *g*, 4 °C), and the supernatant was discarded. The nuclear pellet was re-suspended in nuclear lysis buffer, sonicated, and centrifuged (15 min, 5000 × *g*, 4 °C) to prepare a chromatin solution. The chromatin solution was diluted 10-fold and incubated with control Anti-IgG Rabbit mAb (1:1000, Abcam, catalog no. ab172730, clone no. EPR25A), and Anti-HA-tag Rabbit pAb (1:500, Abcam, catalog no. ab9110) overnight at 4 °C with rotation. DNA was purified from the chromatin solutions, and PCR amplification was performed with primers designed to target the *Pfkl* or *Acadl* promoter region, and the target gel band was assessed. The primers for the sequences of Luciferase and ChIP-PCR assay are: *Pfkl* site1, forward 5'-GGGGTACCGCCTGGGGA

ACCAGGGTTCC-3' and reverse 5'-CCAAGCTTGTACCCGGTTTGTCC CGCCC-3'; *Pfkl* site2, forward 5'-GGGGTACCACAGGCAGGCGCACGGG GCGG-3' and reverse 5'-CCAAGCTTCACCTTGCGCATCACCGCCGCTG-3'; *Pfkl* site3, forward 5'-GGGGTACCATGGGACGCGGGGGCGTGTT TAGGG-3' and reverse 5'-CCAAGCTTGTACGCCAGCTACCCGGGGCGG AGC-3'; *Acadl* site1, forward 5'-GGGGTACCCTGTCACTGAGACTGG GCCG-3' and reverse 5'-CCAAGCTTTAGGCCAAGGAGCTGGTTACC-3'; *Acadl* site2, forward 5'-TCGAGCTAGCATGGCTGCGCGCCTGCTCC TCC-3' and reverse 5'-TCGAGGATCCCTAGCTGTCACTGACGATCTGT-3'; *Acadl* site3, forward 5'-GGGGTACCGCCTGGGGAACCAGGGTTCC-3' and reverse 5'-CCAAGCTTTCAAATTCCGGCCTGTGAAACTT-3'.

### XF24 bioenergetics profiling

The bioenergetics of heart was measured using an Agilent Seahorse XF24 Analyzer. Adult mouse cardiomyocytes (ACMs) were isolated from 3 weeks TG male mice or 8–12 weeks KO male mice and wild-type mice by the lengendorff perfusion method as previously described[49]. The data were processed by Seahorse Wave Controller (version 2.6) software. The Mito Stress Test Kit (Agilent) was used to measure the oxygen consumption rate (OCR). The Glycolytic Rate Assay Kit (Agilent) was used for measuring the extracellular acidification rate (ECAR). On the day prior experimentation, the sensor cartridge was hydrated in a 37 °C non-CO$_2$ incubator and cardiomyocytes seeded at the density of 30,000 ACMs/well or 75,000 NMCMs in to XF24 cell culture microplates and allowed to adhere to the plate overnight. For Palmitate-BSA FAO experiments, etomoxir (40 μM/well) was added at 15 min before OCR analysis, BSA or Palmitate-BSA (175 μM/well) was added to the wells immediately prior to initiate XF assay. Meantime, 1 μM oligomycin, 1 μM 4-trifluoromethoxyphenylhydrazone (FCCP), 0.5 μM rotenone/antimycin A (Rot/AA) for the OCR measurement, 10 mM glucose, 1 μM oligomycin, and 50 mM 2-deoxyglucose (2-DG) for the ECAR measurement were prepared and loaded into the injection ports in the XF24 sensor cartridge. Following microplate insertion, the XF24 protocol consisted of baseline and stepwise injection measurements (3 min for mixture, 2 min for incubation, and 3 min for measurement for a total of three cycles).

### RNA extraction and quantitative PCR

Total RNA was isolated from mouse hearts and NMCMs using TRIzol (Invitrogen, 15596026) according to the manufacturer's protocol. cDNA samples were synthesized with a ReverTra Ace qPCR RT reagent kit (TOYOBO, FSQ-301) according to the manufacturer's instructions, and gene expression was analyzed by qPCR using SYBR Green Master Mix (Roche, 50837000) on ViiA 7 real-time PCR system (ABI) according to the manufacturer's instructions. The primer sequences used in each reaction are listed in Supplementary Table 1.

### High-throughput RNA-seq and data analysis

Cardiac tissues were harvested from KO mice and WT littermates. Ice-cold cardioplegia was perfused into the heart through the cardiac apex

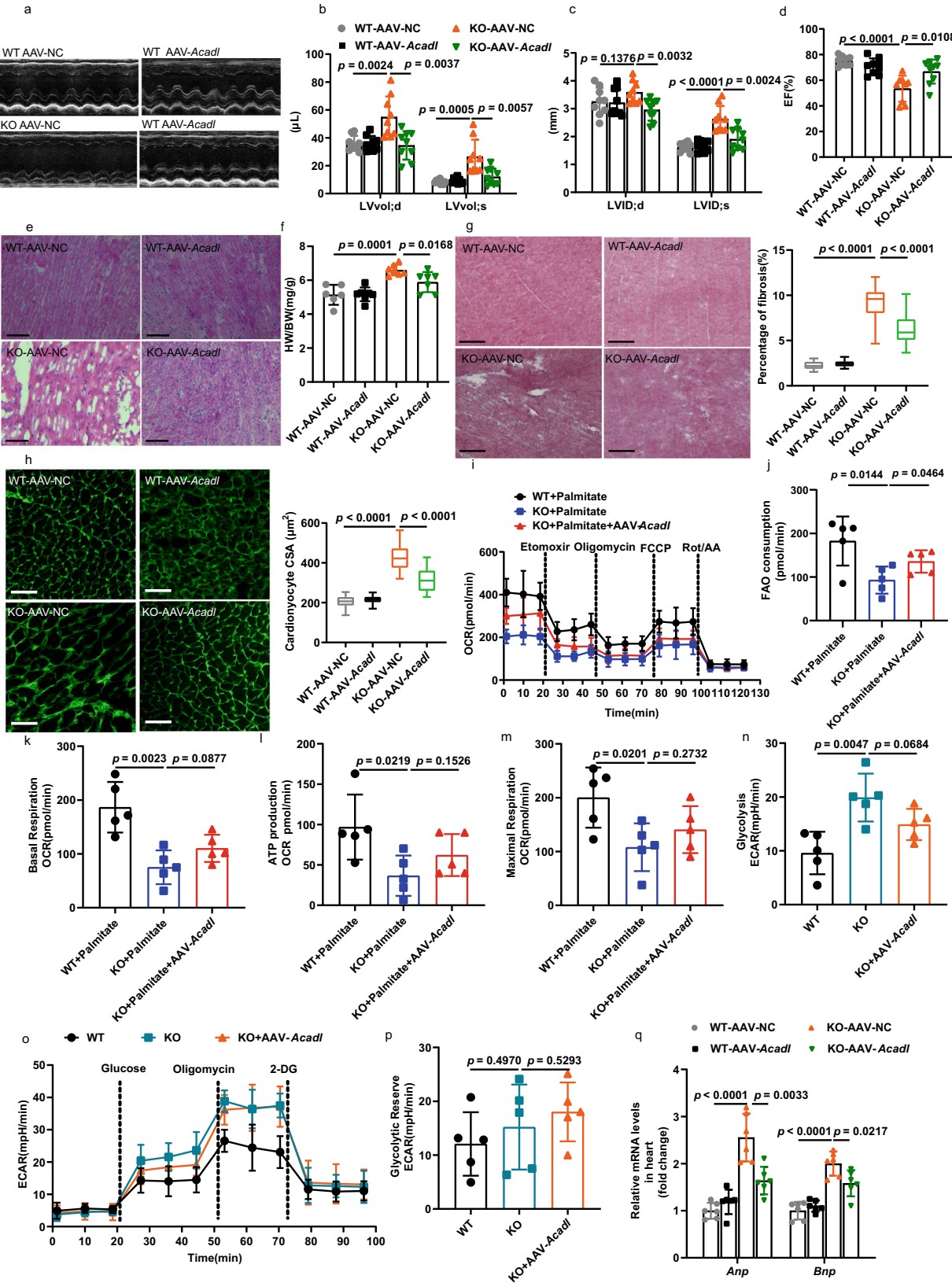

to arrest the heart and prevent ischemic damage. The left ventricle was snap frozen with liquid nitrogen immediately after trimming the atrium, right ventricle and vessels. After an Agilent 2100 bioanalyzer (Agilent RNA 6000 Nano Kit) was used to determine the total RNA sample QC proportion, RNA concentration, RIN value, 28 S/18 S and fragment length distribution, mRNA molecules were isolated from the total RNA

with the oligo (dT) method. Then, the mRNAs were fragmented under specific conditions. First-strand and second-strand cDNA was synthesized, and cDNA fragments were purified and resolved with EB buffer for end repair and single nucleotide (adenine, A) addition. After that, the cDNA fragments were linked to adapters. cDNA fragments of suitable size were selected for PCR amplification. An Agilent 2100 bioanalyser

**Fig. 8 | Cardiac-specific overexpression of *Acadl* protected *Klf7*-deficient hearts from hypertrophic cardiomyopathy. a** Representative echocardiogram of 6-month-old mice treated with AAV-NC or AAV-*Acadl* for 4 weeks. **b–d** Cardiac-specific overexpression of *Acadl* (AAV-*Acadl*) significantly alleviated the increases in LVvol and LVID during systole and diastole and EF induced by cardiac-specific deficiency of *Klf7* in mice (*n* = 9 mice/group). **e** AAV-*Acadl* improved the disordered myocardial tissue arrangement in *Klf7*^KO mice (*n* = 3 mice/group, 8 sections/mouse); scale bar: 100 μm. **f** AAV-***Acadl*** decreased HW/BW in *Klf7*^KO mice (*n* = 6 mice/group). **g** Masson's trichrome staining analysis revealed a reduction in myocardial fibrosis in the *Klf7*^KO mice, (*n* = 3 mice/group, 8 sections/mouse); scale bar: 500 μm. **h** AAV-*Acadl* reduced the enlargement of cardiomyocytes induced by *Klf7* KO, as shown by WGA staining (*n* = 3 mice/group, 8 sections/mouse); scale bar: 100 μm. **i–m** The OCR in cardiomyocytes isolated from mice hearts were measured and

quantification of the FAO consumption and mitochondrial basal respiration, ATP production and maximal respiration rate (*n* = 5 biologically independent samples). **n–p** ECAR in cardiomyocytes isolated from mice hearts were measured at the indicated time points. Quantification of the glycolysis and glycolytic reserve (*n* = 5 biologically independent samples). **q** qRT-PCR analysis showed that AAV-*Acadl* decreased expression of the hypertrophic genes *Anp* and *Bnp* (*n* = 6 biologically independent experiments). **g, h,** The boxplot represents the median shown as a line in the center of the box, the boundaries are the first and third quartile, and whiskers represent the minimum and maximum values in the data. One-way ANOVA with Tukey's multiple comparison test in (**b–d, f–h, j–n, p** and **q**). Data are depicted as the mean values ± SEM. AAV adeno-associated virus, AAV-*Acadl* overexpression *Acadl* with AAV. Source data are provided as a Source Data file.

and ABI StepOnePlus real-time PCR system were used for quantification and qualification of the libraries. We filtered the low-quality reads (more than 20% of the bases showed a quality score lower than 10), then reads with adaptors and reads with unknown bases (more than 5% N bases) to obtain clean reads. Then, we mapped the clean reads onto the reference genome, followed by novel gene prediction, SNP and INDEL calling and gene splicing detection. Finally, we identified DEGs between samples and performed clustering analysis and functional annotation. The MATS statistical model was used to calculate the P-value and false discovery rate (FDR) for the difference in genetic isoform ratio between two conditions. In our project, genes with a FDR < 0.05 were defined as significant differentially spliced genes (DSGs). The data were processed by SOAPnuke (version 1.5.2), HISAT2 (version 2.0.4), StringTie (version 1.0.4), rMATS (version 3.0.9), Bowtie (version 2.2.5) and RSEM (version 1.2.12) software. The analysis was conducted by Shenzhen BGI Technology Co., Ltd.

### AAV9 infection
To achieve *Acadl* overexpression or *Pfkl* knockdown in the hearts of KO mice and WT littermates, AAV9 carrying the mouse *Acadl*-coding gene and AAV9 carrying shRNA to interfere with the *Pfkl* gene were constructed and inserted into the GV571 and GV683 vectors, respectively, and AAV9 encoding GFP was used as a control. AAV particles were used to package constructs and envelope plasmids into 293 A cells. After 48 h of transfection, the cell supernatants were harvested, and the viral particles were concentrated by ultracentrifugation. The viral stocks were re-suspended in serum-free culture medium and stored at −80 °C until use. For myocardial-specific expression, KO mice received AAV9.sh*Pfkl*, AAV9.*Acadl* or NC at 3 E + 11 viral genomes by tail-vein injection. The above AAV was purchased from Shanghai Genechem Co., Ltd (AAV9.sh *Pfkl*, catalog no. GIDV0265700; AAV9.*Acadl*, catalog no. GOSV0285533).

### Targeted metabolomics analysis using mass spectrometry
Approximately 50 mg of frozen tissue was plunged into a 1.5 ml EP tube, an extraction solution was added, and the mixture was vortexed. Then, the sample was ground, subjected to an ultrasonic ice water bath, and centrifuged at 4 °C for 15 min, after which the supernatant in the EP tube was removed and spun dry in a vacuum. Pure water was added for reconstitution, after which the solution was passed through a membrane, and the supernatant was transferred to an LC sample bottle for HPIC-MS/MS analysis.

Mass spectrometry data acquisition and quantitative analysis of target compounds were performed using AB SCIEX Analyst Workstation software, MultiQuant software and Chromeleon 7. The lowest limits of detection (LLODs) for the target compounds were between 0.1-20 nmol/L, and the lowest limits of quantification (LLOQs) were between 10-60 nmol/L. The correlation coefficient (R2) for all target compounds was greater than 0.990, indicating a good quantitative relationship between the chromatographic peak area and the compound concentration, and the average recovery rate for all target

compounds was between 80%-120%, with the standard relative deviation less than 30%. Samples that adhere to these criteria met the requirements of targeted metabolomics analysis. Metabolites were identified by automated comparison of the ion features in the experimental samples to a reference library. The analysis was conducted by the Metabolomics Facility of the Shanghai Baiqu Biomedical Technology Co., Ltd.

### Fluorescence-activated cell sorting
Cardiac cell suspensions were sorted by flow cytometry (BD, FACSAria Fusion) into populations of CD45-CD31+ endothelial cells, CD45-PDGFR-α+ cardiac fibroblasts, CD45-PDGFR-α- CD31- cardiomyocytes, and CD45+F4/80+ macrophages. Isolated cardiac cells were utilized for RNA extraction and qRT-PCR analysis. The antibodies involved including Anti-CD45 mouse mAb (1:1000, BD Pharmingen, catalog no. 553079, clone no. 30-F11), Anti-CD31 mouse mAb (1:1000, BD Pharmingen, catalog no. 558738, clone no. 390), Anti-PDGFR-α mouse mAb (1:1000, BD Pharmingen, catalog no. 558774, clone no. APA5) and Anti-F4/80 mouse mAb (1:1000, BD Pharmingen, catalog no. 567893, clone no. T45-2342). FlowJo (version 7.6.5) was used for flow data analysis.

### U-$^{13}$C-glucose and U-$^{13}$C-palmitate stable isotope tracing
For in vivo tracer analysis, U-$^{13}$C-glucose and U-$^{13}$C-palmitate (1 mg/kg body weight, Cambridge Isotope Laboratories, catalog no. CLM-1396, catalog no. CLM-8390-1) was injected intraperitoneally after 8-12 h fast. After twenty minutes, KO or TG mouse ventricles were excised, washed in ice-cold saline, and immediately frozen in liquid nitrogen. MS was conducted by the Metabolomics Facility of the Shanghai Baiqu Biomedical Technology Co., Ltd.

### Statistical analyses
All data are presented as the mean ± standard error of the mean (SEM). For statistical analysis GraphPad Prism software (version 8.0), Image J (version 2) and Vevo LAB (version 3.2) were used. The exact number of biological replicates (number of mice, samples, or cell culture dishes) is indicated in the figure legends. A two-tailed independent sample *t* test was used to compare differences in mean values between two groups. One-way or two-way ANOVA was used to compare the means of three or more groups. If ANOVA showed a significant difference, then the least significant difference test or Tukey's multiple comparison test was applied for post hoc analysis to detect pairwise differences while adjusting for multiplicity, and the exact *p* values were labeled in the figures.

### Reporting summary
Further information on research design is available in the Nature Portfolio Reporting Summary linked to this article.

## Data availability
The RNA-seq data generated in this study have been deposited in the Zenodo database under accession code zenodo.5525482. The targeted

metabolome sequencing data generated in this study have been deposited in the Zenodo database under accession code zenodo.5525525. The ChIP-seq data generated in this study have been deposited in the Zenodo database under accession code zenodo.5243430. All the data supporting the findings of this study are available in the article, the supplementary information and from the corresponding authors upon request. Source data are provided with this paper.

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

## Acknowledgements

This work was supported by the China National Key R&D Program (project number 2022YFA1604502, recevied by T.W.M.), the National Natural Science Foundation of China (project number 51773050 and 51903067, recevied by T.W.M.) and Heilongjiang Touyan Team (project number HITTY-20190034, recevied by T.W.M.). We thank Professor Zhiwei Huang (School of Life Science and Technology, Harbin Institute of Technology) for disscussion and suggestion on the manuscript. Language editing of this study was provided by Spring Nature Author Services. Figures 2e, 3a, 5a, Supplementary Fig. 5m and Supplementary Fig. 9m were created with BioRender.com.

## Author contributions

T.W.M. and W.C. designed the experiments and supervised the project. W.C., Q.S.P, Z.Y.F., T.H., W.R.Q., Y.W., H.X.L., Z.F.X., and Y.C.F. performed most of biochemical experiments. W.C., Z.B.S., J.J.M., J.Y.W., Z.F.X., and C.Y. performed animal experiments. W.C. and T.W.M. wrote and edited the manuscript.

## Competing interests

The authors declare no competing interests.
