## [Peer Review File · Nature Communications]

The KLF7/PFKL/ACADL axis modulates cardiac metabolic remodelling during cardiac hypertrophy in male miceREVIEWER COMMENTS

Reviewer #1 (Remarks to the Author):

In this paper the authors advance the view that KLF7 inhibits glycolysis and augments FAO. Mice lacking cardiac KLF7 exhibit concentric hypertrophy, increased glycolysis, reduced FAO while mice overexpressing exhibit antiparallel effects (eccentric hypertrophy, reduced glycolysis, and enhanced FAO). Finally, cardiac specific manipulation of a key glycolytic or FAO enzyme partially rescued the effect. Collectively the authors conclude that a KLF7/PFKL/ACADL pathway regulates cardiac hypertrophy.

Overall the studies appear to be well executed and convincing. My major criticism is that the authors have neglected to discuss the rather significant literature on the role of various KLFs in control of cardiac hypertrophy cardiac metabolism (e.g. KLF4, KLF5, KLF10, KLF15). Placing the current work in the context of the field is proper and failure to cite prior work disappointing.

Reviewer #2 (Remarks to the Author):

This manuscript addresses the role of KLF7 in driving the metabolic shift from fat metabolism towards greater dependence on glucose metabolism in cardiac hypertrophy. The authors demonstrate that KLF7 suppresses the expression of PFKL and induces the expression of LCAD. During cardiac hypertrophy KLF7 expression is decreased allowing for PFKL to increase and LCAD to decrease, driving the shift towards more anaerobic metabolism.

Identifying KLF7 as a metabolic regulator relevant to heart failure is a noteworthy finding, that opens up our understanding of the interplay between growth and metabolism.

However, I have a number of technical concerns:

1. The n number for the vast majority of experiments is 3. There is no inclusion of power calculations to justify why such a small sample number is acceptable, given that most studies of this nature use n = 5-6 as a minimum.
2. Changes in metabolism are determined by looking at 1 gene in each pathway (PFK in glycolysis and LCAD in fatty acid oxidation) and metabolomics analysis. This is not sufficient to draw conclusions on cardiac metabolism.
3. The blots for ANP and BNP in Fig 2a look strange - is there any reason why they don't dip downwards like the other protein images?
4. Fig 3 - the lipids that are changing are the very long chain fatty acids - which are metabolised by the peroxisomes not the mitochondria. Why would this be?
5. Lines 84-89 are not accurate reflections of the state of cardiac metabolic research - this comment would have been valid 20 years ago but isn't now.
6. Fig 2 - how much does LCAD go down in panel D - data showing this isn't included.
7. Many of the graphs are so small it was really hard to read them - Figure 5 c was unreadable.

Reviewer #3 (Remarks to the Author):

This study examined the role of Kruppel Like Factor 7 (KLF7) in regulating the expression of key enzymes in glycolysis and fatty acid oxidation and how this contributes to the progression of pathological myocardial remodeling. The authors showed that the downregulation of KLF7 in mice hearts subjected to pressure overload hypertrophy and AngII induced cardiac hypertrophy was accompanied by the increased expression of its target, phosphofructose kinase (PFKL), and

decreased expression of long chain acyl CoA dehydrogenase (ACADL). Cardiac-specific knockout of KLF7 promoted PFKL expression and inhibited ACADL expression, which increased glycolysis and decreased fatty acid oxidation in mouse hearts, while inducing concentric hypertrophy and heart failure. Conversely, cardiac-specific overexpression of KLF7 inhibited PFKL expression and promoted ACADL expression, leading to the significant inhibition of glycolysis and an increase in fatty acid oxidation in the fetal heart, contributing to eccentric cardiac hypertrophy. Cardiac-specific knockdown of PFKL or overexpression of ACADL partially rescued the cardiac hypertrophy phenotype in adult KLF7 deficient mice. It is concluded that the KLF7/PFKL/ACADL metabolic axis is a critical regulatory mechanism and may provide insight into viable therapeutic concepts aimed at the modulation of cardiac metabolic balance in the hypertrophied and failing heart.

General Comments:

This is an interesting study that provides evidence that KLF7 is important in controlling PFKL and ACADL expression. By modifying the expression of KLF7, PFKL or ACADL, the authors also show that this impacts to severity of hypertrophy and heart failure. However, while potentially important, no insights are provided as to why or how inhibiting glycolysis and/or stimulating fatty acid β -oxidation impacts hypertrophy and heart failure.

A second concern with this study is with the indirect way that glycolysis and fatty acid β -oxidation were assessed. Altered flux through these pathways is largely assumed due to differences in expression of either PFKL or ACADL, and in some cases the measurements of various glycolytic or fatty acid β -oxidation intermediates. Neither is a good indicator of actual glycolytic or fatty acid β -oxidative rates.

A third concern is that while a number of different mouse models are used in which KLF7, PFKL or ACADL expression is altered, the authors are not consistent in providing full data sets for each of these experimental groups. These concerns are further elaborated in the Specific Comments.

Specific Comments:

- 1) The bar graphs in the different figures should show the individual data points.
- 2) Figure 3: What does KLF-7 knockdown in NMCs do to PFKL and ACADL. Also, what did knocking down ACADL do to KLF-7 expression or PFKL expression, or increasing PFKL expression do to KLF-7 or ACADL expression. I could not find this data. This data would help solidify the conclusion of the close link between KLF-7, PFKL and ACADL.
- 3) All measures of metabolic flux are indirect, and involve just looking at glycolytic or fatty acid intermediates. In many groups (i.e. PFKL knockdown or ACADL overexpression), this data is missing.
- 4) The authors state that "Metabolomics analysis showed a significant decrease in glycolytic intermediates by the suppression PFKL upon KLF7 overexpression, and metabolic profiling also showed free fatty acid levels to be decreased, indicating that overexpression of KLF7 increased FAO capacity by activating ACADL, most likely impairing cardiac energetics in the fetus (Fig. 5c and 5d, Fig. 7l and 7m in the Supplemental Data)." It is not clear what is meant by "impairing cardiac energetics in the fetus", nor is data provided to support this.
- 5) Figure 6: Overexpression of KLF-7 in the fetus also caused contractile failure. What happens to fatty acid β -oxidation and glycolysis in these hearts?
- 6) The authors state that "Above all, the cardiac-specific overexpression of KLF7 disturbed the expression of enzymes involved in glucose and fatty acid metabolism during the fetal stage, thereby inhibiting the main source of energy during this period (glycolysis), resulting in eccentric cardiac hypertrophy." However, I could not find the data on PFKL, ACADL and glycolytic and fatty acid oxidative enzyme. Why wasn't fatty acid oxidation increased during this period, especially if KLF-7 is increasing fatty acid β -oxidation as proposed?
- 7) Figure 7: Where is the data showing what PFKL knockdown does to ACADL and fatty acid β -oxidation?
- 8) Figure 7: What did PFKL knockdown do to %EF in mice? This data is shown for ACADL

overexpression (Figure 8), why not PFKL knockdown mice? What happens to glycolytic metabolites and fatty acid metabolites in these two sets of mice??

9) The authors focus in on PFKL and ACADL control by KLF7. However, it is not clear whether KLF-7 impacts other fatty acid oxidative enzymes or glycolytic enzymes. Is there any evidence that KLF7 modifies these other enzymes?

10) The authors state that "Conversely, cardiac-specific overexpression of KLF7 inhibited PFKL expression and promoted ACADL expression, leading to the significant inhibition of glycolysis and an increase in fatty acid oxidation in the fetal heart, contributing to eccentric cardiac hypertrophy." These opposing effects in fetal versus adult hearts is confusing.

Reply to Reviewer #1

Comments from Reviewer #1: In this paper the authors advance the view that KLF7 inhibits glycolysis and augments FAO. Mice lacking cardiac KLF7 exhibit concentric hypertrophy, increased glycolysis, reduced FAO while mice overexpressing exhibit antiparallel effects (eccentric hypertrophy, reduced glycolysis, and enhanced FAO). Finally, cardiac specific manipulation of a key glycolytic or FAO enzyme partially reduced the effect. Collectively the authors conclude that a KLF7/PFKL/ACADL pathway regulates cardiac hypertrophy.

1. Overall the studies appear to be well executed and convincing. My major criticism is that the authors have neglected to discuss the rather significant literature on the role of various KLFs in control of cardiac hypertrophy cardiac metabolism (e.g. KLF4, KLF5, KLF10, KLF15). Placing the current work in the context of the field is proper and failure to cite prior work disappointing.

Author response: Thank you for your valuable suggestion. We highly appreciate your comments. According to your suggestion, we have cited the several articles when introducing KLFs involved in cardiac hypertrophy and cardiac metabolism in the revised manuscript. **The following description has been added in the introduction part of the revised manuscript and marked in red.**

Krüppel-like factors (KLFs) are a subfamily of the zinc-finger class of transcriptional regulators. Members of this gene family have been shown to play important roles in cardiac hypertrophy and cardiac metabolism. Previous works have highlighted KLF15 deficient mice develop severe cardiac hypertrophy and heart failure under pressure

overload²⁵⁻²⁷, KLF15 is a direct and independent regulator of myocardial lipid flux and systemic metabolic homeostasis²⁸⁻²⁹. Similarly, KLF10 deficiency results in spontaneous pathological cardiac hypertrophy in mice at the age of 16 months³⁰⁻³¹. KLF5 was shown to be a mediator of cardiac hypertrophy and fibrosis³². A group of studies have shown that KLF4 reactivated fetal cardiac genes during the development of cardiac hypertrophy in vivo³³⁻³⁵, and KLF4 as central for transcriptional control of metabolic function and mitochondrial life cycle in the heart³⁶.

Representative studies regarding this question are listed here as follows:

- [25] Noack, C., *et al.* KLF15-Wnt-Dependent Cardiac Reprogramming Up-Regulates SHISA3 in the Mammalian Heart. *J Am Coll Cardiol* **74**, 1804-1819 (2019).
- [26] Haldar, S. M., *et al.* Klf15 Deficiency Is a Molecular Link Between Heart Failure and Aortic Aneurysm Formation. *Science Translational Medicine* **2**, 26ra26 (2010).
- [27] Fisch, S., *et al.* Kruppel-like factor 15 is a regulator of cardiomyocyte hypertrophy. *PNAS* **104**, 7074-7079 (2007).
- [28] Prosdocimo, D. A., *et al.* Kruppel-like Factor 15 Is a Critical Regulator of Cardiac Lipid Metabolism. *J Biol Chem* **9**, 5914-5924 (2014).
- [29] Fan, L., *et al.* Muscle Krüppel-like factor 15 regulates lipid flux and systemic metabolic homeostasis. *J Clin Invest* **4**, e139496 (2021).
- [30] Rajamannan, N. M., *et al.* TGF β inducible early gene-1 (TIEG1) and cardiac hypertrophy: Discovery and characterization of a novel signaling pathway. *J Cell Biochem* **2**, 315-325 (2007).
- [31] Wang, J., *et al.* Targeted disruption of Smad4 in cardiomyocytes results in cardiac

- hypertrophy and heart failure. *Circ Res* **97**,821-828 (2005).
- [32] Shindo, T., *et al.* Krüppel-like zinc-finger transcription factor KLF5/BTEB2 is a target for angiotensin II signaling and an essential regulator of cardiovascular remodeling. *Nature Medicine* **8**, 856-863 (2002).
- [33] Liao, X. D., *et al.* Krüppel-like factor 4 regulates pressure-induced cardiac hypertrophy. *Journal of Molecular & Cellular Cardiology* **49**, 334-338 (2010).
- [34] Yoshida, T., Kaestner, K. H. and Owens, G. K. Conditional Deletion of Krüppel-Like Factor 4 Delays Downregulation of Smooth Muscle Cell Differentiation Markers but Accelerates Neointimal Formation Following Vascular Injury. *Circ Res* **102**, 1548-1557 (2008).
- [35] Yoshida, T, Yamashita, M, Horimai, C and Hayashi, M. Kruppel-like Factor 4 Protein Regulates Isoproterenol-induced Cardiac Hypertrophy by Modulating Myocardin Expression and Activity. *Journal of Biological Chemistry* **289**, 26107-26118 (2014).
- [36] Liao, X.D., *et al.* Kruppel-like factor 4 is critical for transcriptional control of cardiac mitochondrial homeostasis. *J Clin Invest* **9**, 3461-3476 (2015).

Reply to Reviewer #2

Comments from Reviewer #2: This manuscript addresses the role of KLF7 in driving the metabolic shift from fat metabolism towards greater dependence on glucose metabolism in cardiac hypertrophy. The authors demonstrate that KLF7 suppresses the expression of PFKL and induces the expression of LCAD. During cardiac hypertrophy KLF7 expression is decreased allowing for PFKL to increase and LCAD to decrease,

driving the shift towards more anaerobic metabolism.

Identifying KLF7 as a metabolic regulator relevant to heart failure is a noteworthy finding that opens up our understanding of the interplay between growth and metabolism.

However, I have a number of technical concerns:

1. The n number for the vast majority of experiments is 3. There is no inclusion of power calculations to justify why such a small sample number is acceptable, given that most studies of this nature use n = 5-6 as a minimum.

Author response: Thank you for your generous suggestion. Throughout the text, we have repeated the experiments and increased the number of biological replicates (number of mice, samples, or cell culture dishes) indicated in the figure legends, most experiments were investigated in n = 5-15.

2. Changes in metabolism are determined by looking at 1 gene in each pathway (PFK in glycolysis and LCAD in fatty acid oxidation) and metabolomics analysis. This is not sufficient to draw conclusions on cardiac metabolism.

Author response: Thank you for your valuable suggestion. First of all, we have measured the effect of myocardial-specific overexpression and knockout of KLF7 on the expression of other fatty acid oxidative or glycolytic enzymes, indicated that variation of KLF7 disrupted the expression of FAO enzymes (such as Cpt1b, MCEE, Hadh and ADH1) and glycolytic enzymes (such as HK2, Aldoa, Eno1 and G6PDx) in the heart (**Supplemental Data 9k in the revised manuscript**). Therefore, KLF7

affected the expression of key enzymes involved in cardiac glucose and fatty acid metabolism, resulting in damaging the cardiac energy metabolism. Then, combining the significant differences in the expression of the enzymes mentioned above and the results of ChIP-seq analysis in our study, we currently selected PFKL and ACADL for related research. In addition, PFKL is the most important rate-limiting enzyme of glycolysis, the activation of PFKL has a strong effect on glycolysis, thereby enhancing Warburg effect in tumor cells¹. LCAD catalyzes the initial step in mitochondrial fatty acid oxidation². LCAD knockout mice compared with WT mice, myocardial lipid metabolites accumulated and developed cardiac hypertrophy³. And a study reported that exercise increased glycolysis via phosphorylation of PFK1, and this metabolic shift induced physiological cardiac hypertrophy⁴. The above indicates that PFKL or ACADL differential expression can affect glycolysis and FAO capacity, respectively. Finally, in the revised manuscript, we have supplemented figures of knocking down PFKL and overexpression LCAD in cardiomyocytes to verify the effects of the above two enzymes on glycolysis and fatty acid oxidation fluxes, respectively. As shown in **Supplemental Data 7a-7h in the revised manuscript**, knockdown of PFKL decreased the metabolic flux of glycolysis, whereas overexpression of LCAD increased the metabolic flux of fatty acid oxidation in cardiomyocytes. Taken together, the changes in PFKL and LCAD gene expression involved in our study can influence glycolysis and fatty acid oxidation in the heart.

The following description has been added in the revised manuscript and marked in red.

In addition, a thorough analysis of metabolic enzymes in the KO and TG mice revealed that variation of KLF7 disturbed the balance between cardiac glucose and lipid metabolism (**Supplementary Data 9k in the revised manuscript**).

We evaluated glycolysis and glycolytic reserve capacity by measuring extracellular acidification rate (ECAR), under conditions where NCMs were supplied sequentially with glucose, oligomycin and 2-DG (**Supplemental Data 7a** in the revised manuscript), knocking down of PFKL displayed reduced glycolysis and glycolytic reserve capacity compared to control cardiomyocytes (**Supplemental Data 7a-7c** in the revised manuscript). Meanwhile, we carried out real-time respirometry in cardiomyocytes using BSA as the substrate to assess ACADL mediated fatty acid oxidation (**Supplemental Data 7d-7h** in the revised manuscript). As the data showed, we observed increased fatty acid oxidation consumption (**Supplemental Data 7e** in the revised manuscript), there were no such increase in basal respiration as well as ATP production and maximal respiration between overexpression of ACADL and controls (**Supplemental Data 7f-7h** in the revised manuscript).

k

Supplementary Data 9 (k) The expression of enzymes involved in disordered cardiac glucose and lipid metabolism in TG and KO mouse hearts identified by qRT-PCR (n = 5-6 independent experiments).

Supplementary Data 7 Knocking down PFKL and overexpression of ACADL

inhibited glycolysis and promoted FAO in cultured NMCs, respectively. NMCs stably transfected with control lentivirus (sh-NC) or knocking down PFKFB3 lentivirus (sh-PFKFB3) were supplied with 50 mM glucose, 1 μ M oligomycin and 10 mM 2-DG at the indicated times. ECAR was examined using Seahorse XF24 analyzer (**a**). Relative glycolysis levels (**b**) and glycolytic reserve capacity (**c**). OCR over time using BSA as the substrate in the control and overexpression of ACADL treated cultured cardiomyocytes (**d**), FAO consumption (**e**), Basal respiration (**f**), ATP production (**g**) and maximal respiration (**h**).

Representative studies regarding this question are listed here as follows:

- [1] Li, L., et al. TAp73-induced phosphofructokinase-1 transcription promotes the Warburg effect and enhances cell proliferation. *Nat Commun* **9**: 4683 (2018).
- [2] Kurtz, D. M., et al. Targeted disruption of mouse long-chain acyl-CoA dehydrogenase gene reveals crucial roles for fatty acid oxidation. *Proc Natl Acad Sci U S A* **95**, 15592-15597 (1998).
- [3] Bakermans, A. J., et al. Fasting-induced myocardial lipid accumulation in long-chain acyl-CoA dehydrogenase knockout mice is accompanied by impaired left ventricular function. *Circ: Cardiovasc Imaging* **4**, 558-565 (2011).
- [4] Andrew, A. G., et al. Exercise-Induced Changes in Glucose Metabolism Promote Physiological Cardiac Growth. *Circulation* **136**, 2144-2157 (2017).

3. The blots for ANP and BNP in Fig 2a look strange – is there any reason why they don't dip downwards like the other protein images?

Author response: Thank you for your critical and helpful suggestion. We repeated the mentioned experiment and increased the number of sample. As shown in the follow **Fig. 2a in the revised manuscript**, the shapes of the bands of ANP and BNP were partially corrected.

4. Fig 3- the lipids that are changing are the very long chain fatty acids – which are metabolized by the peroxisomes not the mitochondria. Why would this be?

Author response: Thank you for your valuable suggestion. We highly appreciate your comments. As shown in **Fig. 3**, the free fatty acids we examined include medium-chain FA (6-12 carbon atoms), long-chain FA (14-18 carbon atoms) and very long-chain FA (>18 carbon atoms)¹, and as you mentioned, our results showed that the concentration of most very long-chain FAs changed significantly between KO and WT myocardial tissues, indicated that KLF7 may affect peroxisomes fatty acid metabolism, and we verified that KLF7 can target and negatively regulate Ech1 distributed in peroxisomes (Data not shown). We will further explore this part of content in subsequent studies. Meanwhile, there were also significant differences in the concentration of some medium and long chain FAs (such as C7:0, C14:1, C15:1 and C16:1), and we used seahorse analyzer to further verify the FAO capacity in mitochondria. The results showed that the myocardial-specific knockout of KLF7 mice reduced the fatty acid

oxidation capacity, while the myocardial-specific overexpression of KLF7 enhanced fatty acid oxidation (**Fig. 3f, 3g and Fig. 5f, 5g in the revised manuscript**). Accordingly, we combined the results of the concentration of free fatty acids in myocardial tissue by mass spectrometry and the results of Seahorse detecting fatty acid oxidation capacity of cardiomyocytes isolated from mouse hearts to verify that KLF7 regulated the expression of LCAD and thus affected cardiac fatty acid oxidation.

The following description has been added in the revised manuscript and marked in red.

To directly determine the capacity of FAO, we examined the palmitate based oxygen consumption rate (OCR). Compared to WT mice, an obvious decrease in FAO consumption and respiratory capacity of mitochondria in cardiomyocytes isolated from KO mice (**Fig. 3f, 3g in the revised manuscript**). Similar results were obtained in cardiomyocytes treated with knocking down KLF7 and controls (**Supplemental Data 6d-6h in the revised manuscript**).

In addition, TG mice exhibited a significant rise in FAO, as reflected by the FAO consumption, although there is no significant difference in basal respiration, ATP production and maximal respiration (**Fig. 5f, 5g in the revised manuscript**). Myocardial-specific overexpression of KLF7 in cardiomyocytes promoted the capacity of FAO, this was accompanied by a rise in ATP production and maximal respiration (**Supplemental Data 6l-6p in the revised manuscript**). These results in combination with the metabolomics analysis indicated overexpression KLF7 could affect the capacity of glycolysis FAO in the infant mice heart.

Figure 3 (f) Oxygen consumption rate (OCR) was measured by seahorse analysis with the indicated reagents in cardiomyocytes isolated from WT and KO adult mice. (g) FAO consumption, Basal respiration, ATP production and Maximal respiration were decreased in KO mice compared to WT mice ($n = 5$ technical repeats). The glycolysis and FAO activity were analyzed by the seahorse analyzer as described in the **Methods**.

Figure 5 (f) OCR was measured by seahorse analysis in cardiomyocytes isolated from WT and TG 3-weeks mice. (g) FAO consumption, Basal respiration, ATP production

and Maximal respiration in WT and TG mice (n = 5 technical repeats).

Supplemental Data 6 (d) OCR from a representative experiment where each point of 5 replicates in knocking down KLF7 with lentivirus (sh-KLF7) and controls (sh-NC). **(e)** FAO consumption. **(f)** Basal respiration. **(g)** ATP production. **(h)** Maximal respiration in NMCMs treated with sh-KLF7 and sh-NC. **(m)** OCR from a representative experiment where each point of 5 replicates in overexpression of KLF7 with lentivirus (lenti-KLF7) and controls (lenti-NC). **(i-k)** Overexpression of KLF7 in NMCMs have reduced glycolysis and glycolytic reserve (n = 5 technical repeats) **(o)** FAO consumption. **(l)** Basal respiration. **(n)** ATP production. **(p)** Maximal respiration in NMCMs treated with lenti-KLF7 increased compared to lenti-NC (n = 5 technical repeats).

Representative studies regarding this question are listed here as follows:

[1] Cheon, E. and Mattes R. D., Perceptual Quality of Nonesterified Fatty Acids Varies with Fatty Acid Chain Length. *Chemical Senses* **46**, 1-7 (2021).

5. Lines 84 – 89 are not accurate reflections of the stage of cardiac metabolic research – this comment would have been valid 20 years ago but isn't now.

Author response: Thank you for your valuable suggestion. We highly appreciate your comments. Therefore, we completely deleted lines 84-89 (originally submitted manuscript) and revised it in the revised manuscript.

6. Fig 2 – how much does LCAD go down in panel D – data showing this isn't included.

Author response: Thank you for your generous suggestion. We have supplemented the knockdown efficiency of ACADL in cardiomyocytes (**Supplemental Data 3I in the revised manuscript**), as shown in the follow figure, approximately 40% knockdown of ACADL3 resulted in the reactivation of cardiomyocyte hypertrophy markers compared with the controls.

7. Many of the graphs are so small it was really hard to read them – Figure 5 c was unreadable.

Author response: Thank you for your helpful comments. We replaced all figures with

high resolution in the revised manuscript. In addition, we also carefully rearranged the orders of figures and supplemented the related results in the revised manuscript. We consider that this revision further improves the quality of manuscript by providing what are the scientifically significant findings in the present study. We hope this revision will satisfy your requirements and suggestion.

Reply to Reviewer #3

Comments from Reviewer #3: This study examined the role of Kruppel Like Factor 7 (KLF7) in regulating the expression of key enzymes in glycolysis and fatty acid oxidation and how this contributes to the progression of pathological myocardial remodeling. The authors showed that the downregulation of KLF7 in mice hearts subjected to pressure overload hypertrophy and AngII induced cardiac hypertrophy was accompanied by the increased expression of its target, phosphofructose kinase (PFKL), and decreased expression of long chain acyl CoA dehydrogenase (ACADL). Cardiac-specific knockout of KLF7 promoted PFKL expression and inhibited ACADL expression, which increased glycolysis and decreased fatty acid oxidation in mouse hearts, while inducing concentric hypertrophy and heart failure. Conversely, cardiac-specific overexpression of KLF7 inhibited PFKL expression and promoted ACADL expression, leading to the significant inhibition of glycolysis and an increase in fatty acid oxidation in the fetal heart, contributing to eccentric cardiac hypertrophy. Cardiac-specific knockdown of PFKL or overexpression of ACADL partially rescued the cardiac hypertrophy phenotype in adult KLF7 deficient mice. It is concluded that the KLF7/PFKL/ACADL metabolic axis is a critical regulatory mechanism and may

provide insight into viable therapeutic concepts aimed at the modulation of cardiac metabolic balance in the hypertrophied and failing heart.

General Comments:

This is an interesting study that provides evidence that KLF7 is important in controlling PFKL and ACADL expression. By modifying the expression of KLF7, PFKL or ACADL, the authors also show that this impacts to severity of hypertrophy and heart failure. However, while potentially important, no insights are provided as to why or how inhibiting glycolysis and/or stimulating fatty acid β -oxidation impacts hypertrophy and heart failure.

A second concern with this study is with the indirect way that glycolysis and fatty acid β -oxidation were assessed. Altered flux through these pathways is largely assumed due to differences in expression of either PFKL or ACADL, and in some cases the measurements of various glycolytic or fatty acid β -oxidation intermediates. Neither is a good indicator of actual glycolytic or fatty acid β -oxidative rates.

A third concern is that while a number of different mouse models are used in which KLF7, PFKL or ACAD expression is altered, the authors are not consistent in providing full data sets for each of these experimental groups. These concerns are further elaborated in the Specific Comments.

Specific Comments:

1. The bar graphs in the different figures should show the individual data points.

Author response: Thank you for your valuable suggestion. We have replaced bar graphs with plots that feature information about the distribution of the individual data

points. In addition, all data points have shown for plots with a sample size less than 10, for larger sample sizes, violin plots as alternatives.

2. Figure 3: What does KLF-7 knockdown in NCMs do to PFKL and ACADL. Also, what did knocking down ACADL do to KLF-7 expression or PFKL expression, or increasing PFKL expression do to KLF-7 or ACADL expression. I could not find this data. This data would help solidify the conclusion of the close link between KLF-7, PFKL and ACADL.

Author response: Thank you for your helpful comments. Regarding your suggestion, we have supplemented data on the effect of knocking down ACADL on KLF7 and PFKL expression and the effect of overexpressing PFKL on KLF7 and ACADL expression in cardiomyocytes (**Supplemental Data 3h, 3g in the revised manuscript**). As shown in the figure, knockdown of ACADL had no significant effect on the protein expression of KLF7 and PFKL. Similarly, overexpression of PFKL had no significant effect on the expression of ACADL and KLF7. However, knockdown of KLF7 promoted the expression of PFKL and inhibited the expression of ACADL in our study. Our results verified that KLF7 targets PFKL and ACADL, and variation in the expression of PFKL or ACADL had no effect on the expression of the other two.

The following description has been added in the revised manuscript and marked in red.

To fully verify the conclusion of the close link between KLF7, PFKL and ACADL, we observed overexpression of PFKL had no obvious effect on the expression of KLF7 and

ACADL (**Supplemental Data 3g in the revised manuscript**). We also checked KLF7 and PFKL expression in knockdown ACADL treated NMCMs. Our results showed there was no significantly difference between KLF7 and PFKL expression by knockdown ACADL in NMCMs (**Supplemental Data 3h in the revised manuscript**).

Supplemental Data 3 (g) Western blot analysis of KLF7 and PFKL in NMCMs treated with overexpression PFKL (lenti-PFKL) and controls (lenti-NC). **(h)** Protein expression analysis of ACADL, KLF7 and PFKL expression in knockdown ACADL (sh-ACADL) and controls (sh-NC) (n = 4-5 independent experiments).

3. All measures of metabolic flux are indirect, and involve just looking at glycolytic or fatty acid intermediates. In many groups (i.e. PFLK knockdown or ACADL overexpression), this data is missing.

Author response: Thank you for your valuable suggestion. Based on your suggestion, we used Seahorse XF24 analyzer to examine the effects of PFKL and ACADL on glycolysis and FAO, respectively. As shown **Supplemental Data 7a-7h in the revised manuscript**, myocardial-specific knockdown of PFKL in cardiomyocytes inhibited

glycolytic capacity, while overexpression of ACADL promoted FAO. Therefore, we supplemented the above data to more directly validate the evidence that the KLF7/PFKL/ACADL axis affects the pathological process of cardiac hypertrophy by regulating the rate of glycolysis and FAO in the heart.

The following description has been added in the revised manuscript and marked in red.

In addition, we evaluated glycolysis and glycolytic reserve capacity by measuring extracellular acidification rate (ECAR), under conditions where NMCMs were supplied sequentially with glucose, oligomycin and 2-DG (**Supplemental Data 7a**), knocking down of PFKL displayed reduced glycolysis and glycolytic reserve capacity compared to control cardiomyocytes (**Supplemental Data 7b, 7c**). Meanwhile, we carried out real-time respirometry in NMCMs using BSA as the substrate to assess ACADL mediated fatty acid oxidation (**Supplemental Data 7d-7h**). As the data showed, we observed increased fatty acid oxidation consumption (**Supplemental Data 7e**), there were no such increase in basal respiration as well as ATP production and maximal respiration between overexpression of ACADL and controls (**Supplemental Data 7f-7h**).

Supplemental Data 7 Knocking down PFKL and overexpression of ACADL

inhibited glycolysis and promoted FAO in cultured NMCMs, respectively.

NMCMs stably transfected with control lentivirus (sh-NC) or knocking down PFKL lentivirus (sh-PFKL) were supplied with 50 mM glucose, 1 μ M oligomycin and 10 mM 2-DG at the indicated times. ECAR was examined using Seahorse XF24 analyzer (a). Relative glycolysis levels (b) and glycolytic reserve capacity (c). OCR over time using BSA as the substrate in the control and overexpression of ACADL treated cultured cardiomyocytes (d), FAO consumption (e), Basal respiration (f), ATP production (g) and maximal respiration (h).

4. The authors state that “Metabolomics analysis showed a significant decrease in glycolytic intermediates by the suppression PFKL upon KLF7 overexpression, and metabolic profiling also showed free fatty acid levels to be decreased, indicating that overexpression of KLF7 increased FAO capacity by activating ACADL, most likely impairing cardiac energetics in the fetus (Fig. 5c and 5d, Fig. 7l and 7m in

the Supplemental Data).” It is not clear what is meant by “impairing cardiac energetics in the fetus”, nor is data provided to support this.

Author response: Thank you for your critical and helpful comments. Throughout the description of our study and combined with the experimental results, the word “fetus” we used is indeed inappropriate, therefore we have replaced “fetus” with “infant mice” in the revised manuscript.

5. Figure 6: Overexpression of KLF-7 in the fetus also caused contractile failure.

What happens to fatty acid β -oxidation and glycolysis in these hearts?

Author response: Thank you for your valuable suggestion. We used the Seahorse XF24 analyzer to examine the glycolytic and FAO capacities of cardiomyocytes isolated from 3-week-old TG mice and NMCs treated with lentivirus overexpression KLF7, respectively. As shown in **Fig. 5d-5g, Supplemental Data 6i-6p in the revised manuscript**, whether at the cellular or animal level, myocardial-specific overexpression of KLF7 led to a decrease in glycolysis and an increase in FAO, resulting in insufficient cardiac energy supply and cardiac systolic dysfunction in infant mice.

The following description has been added in the revised manuscript and marked in red.

Seahorse analyses further confirmed that cardiomyocytes isolated from TG mice displayed reduced glycolysis compared with WT cardiomyocytes (**Fig. 5d, 5e**). In addition, TG mice exhibited a significant rise in FAO, as reflected by the FAO consumption, although there is no significant difference in basal respiration, ATP

production and maximal respiration (**Fig. 5f, 5g**). More importantly, In vitro level, overexpression KLF7 NMCMs displayed a reduced rate of glycolysis compared with controls (**Fig. 6i-6k in the Supplemental Data**). Myocardial-specific overexpression of KLF7 in cardiomyocytes promoted the capacity of FAO, this was accompanied by a rise in ATP production and maximal respiration (**Fig. 6l-6p in the Supplemental Data**). These results in combination with the metabolomics analysis indicated overexpression KLF7 could affect the capacity of glycolysis and FAO in the infant mice heart.

Figure 5 (d) ECAR of isolated cardiomyocytes using the Seahorse XF24 to analyze 3-weeks TG mice glycolytic function. (e) TG mice have decreased glycolysis and glycolytic reserve activities. (n = 5 technical repeats) (f) OCR was measured by seahorse analysis in cardiomyocytes isolated from WT and TG 3-weeks mice. (g) FAO consumption, Basal respiration, ATP production and Maximal respiration in WT and TG mice (n = 5 technical repeats).

Supplemental Data 6 (i-k) Overexpression of KLF7 in NCMs have reduced glycolysis and glycolytic reserve (n = 5 technical repeats) (**o**) FAO consumption. (**l**) Basal respiration. (**n**) ATP production. (**p**) Maximal respiration in NCMs treated with lenti-KLF7 increased compared to lenti-NC (n = 5 technical repeats).

6. The authors state that “Above all, the cardiac-specific overexpression of KLF7 disturbed the expression of enzymes involved in glucose and fatty acid metabolism during the fetal stage, thereby inhibiting the main source of energy during this period (glycolysis), resulting in eccentric cardiac hypertrophy.” However, I could not find the data on PFKL, ACADL and glycolytic and fatty acid oxidative enzyme. Why wasn’t fatty acid oxidation increased during this period, especially if KLF-7 is increasing fatty acid β -oxidation as proposed?

Author response: Thank you for your helpful suggestion. We have supplemented the expression data of PFKL and ACADL in myocardial tissue of TG and WT mice. First of all, we examined the transcript of PFKL and ACADL in the myocardial tissue in

different ages (**Supplementary Data 9l in the revised manuscript**), and then we selected the time point of 3-week to determine the protein expression of PFKL and ACADL (**Figure 5b, 5c in the revised manuscript**). Finally, the data on the effect of overexpression KLF7 on FAO you mentioned has been elaborated in **Comment 5**, overexpression of KLF7 significantly increased the FAO capacity of the heart.

The following description has been added in the revised manuscript and marked in red.

Compared with their expression in WT mice, overexpression of KLF7 significantly downregulated PFKL expression and upregulated ACADL expression in 3-week TG mice after birth, and ACADL expression was reduced at 12-week, suggesting that the ability of the TG mice to carry out glycolysis and fatty acid metabolism was reduced at 12 weeks, and the mice exhibited a severe HF phenotype at this stage (**Fig. 5b, 5c and Fig. 9l in the Supplemental Data**).

Figure 5 (b-c) The protein level of the KLF7 target gene PFKL was downregulated, and the ACADL protein level was upregulated in myocardial tissues of TG mice of different ages compared to their expression in corresponding WT mice (n = 5-9 independent experiments).

Supplementary Data 9 (I) The mRNA levels of the KLF7 target gene PFKL were downregulated, while those of ACADL were upregulated in the myocardial tissues of TG mice compared to WT mice at different ages (n = 6 independent experiments).

7. Figure 7: Where is the data showing what PFKL knockdown does to ACADL and fatty acid β -oxidation?

Author response: Thank you for your generous suggestion. In vitro, the expression of ACADL and FAO capacity were examined after the NMCs were overexpressed PFKL. As shown in the follow figure, compared with the sh-NC group, the myocardial-specific overexpression of PFKL had no significant effect on the expression of ACADL, and the FAO metabolic flux result showed that the overexpression of PFKL did not impact FAO capacity.

The following description has been added in the revised manuscript and marked in red.

To fully verify the conclusion of the close link between KLF7, PFKL and ACADL, we observed overexpression of PFKL had no obvious effect on the expression of KLF7 and ACADL (**Supplemental Data 3g in the revised manuscript**).

To determine the impact of PFKL downregulation on cardiomyocytes FAO metabolism, we quantified cellular OCR. **Supplementary Data 7i-m** showed the time course of protocol with the injection of each compound and the impact on OCR. The results demonstrated that knockdown PFKL had a tendency to increase in FAO capacity, but there was no significant difference, as evidenced by increased FAO consumption in comparison with controls.

Supplementary Data 3 (g) Western blot analysis of KLF7 and PFKL in NMCMs treated with overexpression PFKL (lenti-PFKL) and controls (lenti-NC). **(h)** Protein expression analysis of ACADL, KLF7 and PFKL expression in knockdown ACADL (sh-ACADL) and controls (sh-NC) (n = 4-5 independent experiments).

Supplementary Data 7 Assessment of OCR in cardiomyocytes treated with

knockdown PFKL and controls. (i) NMCMs were exposed sequentially to Etomoxir, Oligomycin, FCCP and Rot/AA, and OCR was measured over time using a Seahorse XF24 analyzer, cardiomyocytes FAO capacity test was performed according to manufacturer's protocol (n = 5 technical repeats).

8. Figure 7: What did PFKL knockdown do to %EF in mice? This data is shown for ACADL overexpression (Figure 8), why not PFKL knockdown mice? What happens to glycolytic metabolites and fatty acid metabolites in these two sets of mice??

Author response: Thank you for your helpful and critical comment. First of all, as shown in the **Supplementary Data 10d in the revised manuscript**, knockdown of PFKL with adeno-associated virus in WT mice had no significant effect on cardiac ejection fraction (%EF), which is attributed to the fact that cardiac energy metabolism in adult mice mainly depends on FAO, while glycolysis accounts for a small part. Therefore, knockdown of PFKL may have less effect on cardiac energy metabolism in WT mice, resulting in no significant effect on cardiac systolic function. However, myocardial-specific knockout of KLF7 resulted in enhanced glycolytic capacity and reduced FAO in the heart, and knockdown of PFKL attenuated the glycolysis and slightly enhanced FAO capacity in KO mice (**Supplementary Data 7h-7i in the revised manuscript**), which partially restored %EF. Then, the Fig.7 in our revised manuscript determined the validation of myocardial-specific knockdown of PFKL on the restoration of cardiac function and metabolic capacity in KO mice. The central idea

of our study is to explore whether KLF7 has targeted regulation on PFKL and ACADL, and according to your **comment 2**, we have supplemented whether there is close regulatory correlation between PFKL and ACADL, the results showed that there was no direct regulatory correlation (**Supplemental Data 3g in the revised manuscript**), while KLF7 could positively regulate the expression of ACADL and negatively regulate the expression of PFKL. Therefore, we used KLF7 knockout mice to verify the impact of PFKL and ACADL on cardiac function, pathology morphology and metabolic flux. Finally, in the animal rescue experiments, we have supplemented the data on the effects of knockdown PFKL and overexpression ACADL on the glycolysis and FAO capacity in the heart of KO mice, respectively. In conclusion, myocardial-specific knockdown of PFKL or overexpression of ACADL in KO mice partially alleviated the impairment of cardiac energy metabolism caused by knockout KLF7 (**Fig. 7h-7n, Figure 8i-8p in the revised manuscript**).

The following description has been added in the revised manuscript and marked in red.

There was no significant difference in PFKL-knockdown WT mice compared to NC-knockdown WT mice in (%)EF (**Supplementary Data 10d in the revised manuscript**).

To assess whether the observed knockdown PFKL and overexpression ACADL were correlated with glycolysis and FAO function, respectively. Metabolic flux analysis was performed, and the OCR and ECAR were measured in WT, KO and KO-AAV-PFKL and KO-AAV-ACADL mice heart using Seahorse XF24 analyzer. The results showed that knockdown PFKL significantly decreased glycolysis in the KO-AAV-PFKL mice

heart compared with KO mice (**Figure 7h, 7i in the revised manuscript**), however, there was no significant difference between KO and KO-AAV-PFKL mice in FAO capacity (**Figure 7j-7n in the revised manuscript**). In addition, myocardial-specific overexpression ACADL (KO-AAV-ACADL) exhibited enhanced FAO capacity, as reflected by increased FAO consumption (**Figure 8i-8m in the revised manuscript**). Next, we evaluated the ECAR in the overexpression ACADL in KO mice. As shown in the **Figure 8n-8p (in the revised manuscript)**, overexpression ACADL did not impact glycolysis and glycolytic reserve capacity. Above all, these results indicated that myocardial-specific knockout KLF7 impaired cardiac energy metabolism, however, this impairment was markedly blunted in treatment with knockdown PFKL or overexpression ACADL.

Supplementary Data 10 Echocardiographic parameters of the groups: the ejection fraction (EF) (n = 7 mice/group).

Figure 7 (h-i) ECAR in cardiomyocytes isolated from mice hearts were measured under basal, glucose (10 mM), oligomycin (1 μ M) and 2-DG (50 mM) stimulated conditions at the indicated time points. Quantification of the glycolysis and glycolytic reserve (n = 5 technical repeats). **(j)** OCR in cardiomyocytes isolated from mice hearts were measured under basal, Etomoxir (40 μ M), oligomycin (1 μ M), FCCP (1 μ M) and Rot/AA (0.5 μ M) conditions at the indicated time points. **(k-n)** Quantification of the FAO consumption and mitochondrial basal respiration, ATP production and maximal respiration rate (n = 5 technical repeats).

Figure 8 (i-m) The OCR in cardiomyocytes isolated from mice hearts were measured and quantification of the FAO consumption and mitochondrial basal respiration, ATP production and maximal respiration rate (n = 5 technical repeats). **(n-p)** ECAR in cardiomyocytes isolated from mice hearts were measured at the indicated time points. Quantification of the glycolysis and glycolytic reserve (n = 5 technical repeats).

9. The authors focus in on PFKL and ACADL control by KLF7. However, it is not clear whether KLF-7 impacts other fatty acid oxidative enzymes or glycolytic enzymes. Is their any evidence that KFL7 modifies these other enzymes?

Author response: Thank you for your helpful suggestion. We have supplemented the data on the effect of myocardial-specific overexpression and knockout of KLF7 on the

expression of other fatty acid oxidative enzymes or glycolytic enzymes, and variation of KLF7 disrupted the expression of FAO enzymes (such as Cpt1b, MCEE, Hadh and ADH1) and glycolytic enzymes (such as HK2, Aldoa, Eno1 and G6PDx) in the heart (**Supplemental Data 9k in the revised manuscript**), indicated that KLF7 affected the expression of key enzymes involved in cardiac glucose and fatty acid metabolism, resulting in damaging the cardiac energy metabolism. More importantly, combining the significant differences in the expression of the enzymes mentioned above and the results of ChIP-seq analysis in our study, we currently selected PFKL and ACADL for related research.

The following description has been added in the revised manuscript and marked in red.

In addition, a thorough analysis of metabolic genes in the WT and TG mice revealed that KLF7 overexpression disturbed the balance between cardiac glucose and lipid metabolism (**Supplementary Data 9k in the revised manuscript**).

k

Supplementary Data 9 (k) The expression of enzymes involved in disordered cardiac glucose and lipid metabolism in TG and KO mouse hearts identified by qRT-PCR (n = 5-6 independent experiments).

10. The authors state that “Conversely, cardiac-specific overexpression of KLF7 inhibited PFKL expression and promoted ACADL expression, leading to the significant inhibition of glycolysis and an increase in fatty acid oxidation in the fetal heart, contributing to eccentric cardiac hypertrophy.” These opposing effects in fetal versus adult hearts is confusing.

Author response: Thank you for your helpful comments. According to previous studies, dramatic changes occur in cardiac energy metabolism during cardiac development,

differentiation and postnatal growth. During early cardiac development, the heart resides in a low-oxygen environment and is therefore highly dependent on glycolysis (44%) as an ATP-generating pathway, with FAO contributing only a small fraction (13%) of total myocardial ATP production. As cardiomyocytes mature and become terminally differentiated, mitochondrial FAO (70 - 90%) is the main source of energy in the adult cardiomyocytes, and glycolysis (5%) accounts for a small proportion¹⁻². Therefore, in our study, TG mice had genetically stably overexpression KLF7 at the embryonic stage, and KLF7 significantly inhibited the expression of PFKL, the rate-limiting enzyme of glycolysis, contributing to infant TG mice hearts inhibited glycolysis. Overexpression KLF7 slightly increased FAO capacity. However, in the immature stage, the main energy of the heart derives from glycolysis, resulted in TG mice had pathological symptoms of eccentric cardiac hypertrophy. Contrastly, in adult, myocardial-specific knockout KLF7 downregulated ACADL expression and upregulated PFKL expression, the reduced FAO and increased glycolysis capacity have the same metabolic profile as those of concentric cardiac hypertrophy. Therefore, KO adult mice developed symptoms of concentric cardiac hypertrophy and heart failure.

We consider that this revision further improves the quality of manuscript by providing what are the scientifically significant findings in the present study. We hope this revision will satisfy your requirements and suggestion.

Representative studies regarding this question are listed here as follows:

[1] Gibb, A. A. and Hill. B. G. Metabolic Coordination of Physiological and Pathological Cardiac Remodeling. *Circulation Research* **123**, 107-128 (2018).

[2] Lopaschuk, G. D. and Jaswal, J. S. Energy metabolic phenotype of the cardiomyocyte during development, differentiation, and postnatal maturation. *J Cardiovasc Pharmacol* **56**, 130-140 (2010).

REVIEWER COMMENTS

Reviewer #1 (Remarks to the Author):

No further issues.

Reviewer #3 (Remarks to the Author):

The authors have extensively revised this manuscript and have adequately addressed most of the concerns I raised in my initial review of the paper. However, my concern that "no insights are provided as to why or how inhibiting glycolysis and/or stimulating fatty acid oxidation impacts hypertrophy and heart failure" has not been addressed.

Reviewer #4 (Remarks to the Author):

The authors have added further metabolic measurements using the Seahorse analyser in isolated cardiomyocytes. This adds to the paper but falls short of metabolic flux measurements in vivo. This experiment would really prove if there is an altered metabolism in the hearts rather than in isolated cells in culture.

Other main points:

1) In response to reviewer 1, the authors point out that other KLFs also induce cardiac phenotypes - KLF15, KLF10, KLF5 and KLF4. How are they expressed in different cell populations in the heart? How do they change in response to manipulating KLF7?

2) A more fundamental question is relating to your rationale for making a cardiac-specific KLF7 KO when the human single cell RNA seq data seem to show that KLF7 is barely expressed in cardiomyocytes, but predominantly found in endothelial cells? There is also no overlap between KLF7, ACADL and PFKL expression? ACADL does not seem to be detected either?

3) "Fig 1 (a) Analysis of the GEO single-cell sequencing database revealed a decrease in KLF7 expression. (b, c) increase in PFKL expression and decrease in ACADL expression in dilated cardiomyopathy patients (n = 11492 cells). "

This change is not apparent from those figures. All I see is a huge variability of the expression levels.

4) The Western blots are of very poor quality. I would consider all the blots for ACADL invalid - smearing of band makes these readouts not reliable for quantitation.

5) The Western blot in Fig 2b has a MW shift in GAPDH between lanes?! Looking at the raw data in the folder "Western Blot NMCMs+lenti-PFKL" it is apparent that from "GAPDH(BNP).tif" the first four lanes on the original plot were cut and cropped together with the following 5 lanes so they became the last 4 lanes instead. The cropping was done so poorly that it resulted in a shift in GAPDH as well as ANP. Moreover, GAPDH has a MW of 36kD, it's annotated correctly as 37kD in the main figure 2, but in the original plot "Western Blot NMCMs+lenti-PFKL" "GAPDH(BNP).tif" the band is running between 10-25kD instead? The Western blot membranes should be presented in their entirety in the supplement, instead the blots provided are again cropped.

6) Suppl Data 3:

"(h) Protein expression analysis of ACADL, KLF7 and PFKL expression in knockdown ACADL (sh-ACADL) and controls (sh-NC) (n = 4-5 independent experiments)"

Panel H -the n-numbers for sh-NC and sh-ACADL in the Western blot presented are n=5,

densitometry for KLF7 however only shows quantification for n=4 in controls and sh-ACADL group? For PFKL on the other hand, the n-number is n=8-10? How is that possible? Are you referring biological replicates, technical replicates or to independent experiments?

7) Suppl Data 5:

The cardiac specific KLF7 KO reduces KLF7 content in heart by 50%. Again, the Western blot is very poor. Staining for KLF7 in the different cardiac cell types would be important. Confirmation of successful KO in cardiomyocyte could be performed in isolated cardiomyocytes with PCR and compared to the non-myocyte fraction.

8) Suppl Data 6 :

KLF7 knock-down in NMCs. How high is KLF7 expressed in NMCs? How does it compare to adult myocytes? How much is it reduced with sh-KLF7? As mentioned above, what are the effects of manipulating KLF7 on other KLF family members?

Reply to Reviewer #3

Comments from Reviewer #3: The authors have extensively revised this manuscript and have adequately addressed most of the concerns I raised in my initial review of the paper. However, my concern that "no insights are provided as to why or how inhibiting glycolysis and/or stimulating fatty acid oxidation impacts hypertrophy and heart failure" has not been addressed.

Author response: Thank you for your valuable suggestion. We highly appreciate your comments. Alterations in cardiac energy substrate metabolism are recognized as important hallmarks of the cardiac remodeling process. The decrease in FAO is accompanied by a shift to the fetal metabolic profile, and is characterized by an increased reliance on glycolysis for ATP generation⁵⁷. Moreover, several lines of evidence suggest that changes in myocardial fuel utilization are the cause of the hypertrophic growth response and subsequent remodeling⁵⁶⁻⁵⁸. Inhibition of glycolysis and/or stimulation of FAO could alleviate the adverse cardiac remodeling of cardiac hypertrophy and heart failure⁵⁹⁻⁶⁴. Therefore, we designed the idea that KLF7 could simultaneously target and regulate the expression of key enzymes in the glycolysis and fatty acid oxidation metabolic pathways to alleviate cardiac remodeling in cardiac hypertrophy and heart failure. This is why inhibition of glycolysis or stimulation of FAO can have a key role in cardiac hypertrophy and heart failure. More importantly, in our study, we demonstrated at both in vitro and in vivo that knockdown PFKL and overexpression of ACADL partially alleviated the phenotype of cardiac hypertrophic growth induced by knockdown of KLF7 by inhibiting glycolysis and enhancing FAO, respectively. In our follow-up study, we will focus on the effect of small molecule drug targeting KLF7 on cardiac metabolic remodeling, providing a basis for the clinical treatment of cardiac hypertrophy and heart failure. This is how inhibiting glycolysis

and/or stimulating fatty acid oxidation impacts hypertrophy and heart failure.

The following description has been added in the discussion part of the revised manuscript and marked in red.

Thus, the question of a casual role of alterations in myocardial metabolism has been controversial for decades. Several lines of evidence suggest that changes in myocardial fuel utilization are an early event that occurs with the hypertrophic growth response and subsequent remodeling⁵⁶⁻⁵⁸. Moreover, many studies have found that inhibition of glycolysis and/or stimulation of FAO could alleviate the pathological remodeling of cardiac hypertrophy or HF. Targeting the enzymes or genes responsible for, or controlling FAO in the heart could provide novel therapeutic insights for treating HF⁵⁹. Cardiac-specific deletion of ACC2 which results in high cardiac FAO rates, are protected against the development of cardiac hypertrophy⁶⁰. Astragaloside IV switched glycolysis to FAO and improved mitochondrial function, which may present a novel cardio-protective treatment that inhibits the progress of HF⁶¹. The empaglifloxin reduce glycolysis, rebalance coupling between glycolysis and oxidative phosphorylation to attenuate adverse cardiac remodeling and progression of HF induced by pressure-overload⁶². Serpina3c inhibits glycolysis and provides a potential intervention target for treatment of HF⁶³. Inhibition of AMP-activated kinase in vivo prevented enhanced glycolytic flow and rescued cardiac hypertrophy⁶⁴.

Furthermore, in our study, we demonstrated at both in vitro and in vivo that knockdown PFKL and overexpression of ACADL partially alleviated the phenotype of cardiac hypertrophic growth induced by knockdown of KLF7 by inhibiting glycolysis and enhancing FAO, respectively.

Representative studies regarding this question are listed here as follows:

[56] Matsuura, T. R. , Leone, T. C. , Kelly, D. P. Fueling Cardiac Hypertrophy.

- Circulation Research* **2**, 197-199 (2020).
- [57] Zhang, L. , *et al.* Cardiac insulin-resistance and decreased mitochondrial energy production precede the development of systolic heart failure after pressure-overload hypertrophy. *Circulation Heart Failure* **5**, 1039-1048 (2013).
- [58] Torsten, D. , *et al.* Decreased rates of substrate oxidation ex vivo predict the onset of heart failure and contractile dysfunction in rats with pressure overload. *Cardiovascular Research* **3**, 461-470 (2010).
- [59] Arumugam, S. , *et al.* Targeting fatty acid metabolism in heart failure: is it a suitable therapeutic approach? *Drug Discovery Today*, 1003-1008 (2016).
- [60] Kolwicz, S. C. , *et al.* Cardiac-specific deletion of acetyl CoA carboxylase 2 prevents metabolic remodeling during pressure-overload hypertrophy. *Circulation Research* **111**, 728-738 (2012).
- [61] Zhiwei, *et al.* Astragaloside IV alleviates heart failure via activating PPAR α to switch glycolysis to fatty acid β -oxidation. *Scientific Reports* **7**, 2691-2706 (2017).
- [62] Li, X. , *et al.* Direct Cardiac Actions of the Sodium Glucose Co-Transporter 2 Inhibitor Empagliflozin Improve Myocardial Oxidative Phosphorylation and Attenuate Pressure-Overload Heart Failure. *Journal of the American Heart Association* **10.6**:e018298 (2021).
- [63] Ji J. J. , Yao Y . Kallistatin/ serpin3c inhibits fibrosis after myocardial infarction by regulating glycolytic pathway. *European Heart Journal*, (2021).
- [64] Biesemann N , Braun T . Myostatin Regulates Energy Homeostasis in the Heart and Prevents Heart Failure. *Circulation Research* **7**, 296-310 (2014).

Reply to Reviewer #4

Comments from Reviewer #4: The authors have added further metabolic

measurements using the Seahorse analyser in isolated cardiomyocytes. This adds to the paper but falls short of metabolic flux measurements in vivo. This experiment would really proof if there is an altered metabolism in the hearts rather than in isolated cells in culture.

Author response: Thank you for your valuable suggestion. We highly appreciate your comments. We have supplemented the metabolic flux measurements in KO and TG mice. As shown in the following figures, consistent with the results of glycolysis and FAO in isolated cardiomyocytes. We used the metabolic flux analysis method of isotope-labeled glucose [U-¹³C] to determine the metabolic function changes of the glycolytic pathway. Myocardial-specific knockout of KLF7 increased the production rate of labelled glycolytic intermediates, especially F-6-P (m+6), F-1, 6-BP (m+6) and PDE (m+3) were significantly increased (**Figure 3i and 3j in the revised manuscript**). Conversely, TG mice significantly decreased the production rate of labelled glycolytic intermediates, such as F-6-P (m+6), F-1, 6-BP (m+6) and Pyruvate (m+3) (**Figure 5i in the revised manuscript**). Meanwhile, the mice intraperitoneally injected stable isotope labelled palmitate [U-¹³C] was used to calculate the contribution of FAO to acetyl-CoA entering the TAC¹. As shown in the **Figure 3k and Figure 5j in the revised manuscript**, cardiac-specific knockout KLF7 decreased the abundance of the labelled acetyl-CoA (m+2) and TG mice increased the abundance of the labelled acetyl-CoA (m+2). In conclusion, consistent with the in vitro results, the in vivo metabolic flux results suggest that knockout of KLF7 promoted glycolysis inhibited FAO, while overexpression of KLF7 inhibited glycolysis and promoted FAO, thereby disturbing the balance between glucose and fatty acid metabolism in the adult and infant heart, respectively.

The following description has been added in the revised manuscript and marked

in red.

We next conducted a tracer experiment to further determine changes in glycolytic flux in vivo. We treated KO and WT mice with [U-¹³C] glucose and harvested the hearts 20 minutes later. Glycolytic intermediates, such as F-6-P, F-1, 6-BP and PDE, were significantly increased in KO mice (**Figure 3i, 3j**). The abundance of Acetyl-CoA labelled [U-¹³C]-Palmitate was decreased in KO hearts when compared with WT hearts (**Figure 3k**).

As shown in metabolic flux analysis in vivo, glycolytic intermediates, such as F-6-P, F-1, 6-BP and pyruvate, were significantly decreased in TG mice (**Figure 5i**). The abundance of Acetyl-CoA labelled [U-¹³C]-Palmitate was increased in TG hearts when compared with WT hearts (**Figure 5j**).

Figure 3 (i) Schematic of ¹³C-labeled glucose metabolic flux analysis in KO and WT mouse left ventricular tissue. (j) Quantification results of ¹³C-glucose metabolic flux analysis. F-6-P (m+6), F-1, 6-BP (m+6) and PDE (m+3) were increased by knockout KLF7. Results are showed as fractional changes. (n = 3 independent heart samples per

group). (k) The mice intraperitoneal injection stable isotope labelled palmitate was used to calculate the contribution of FAO to acetyl-CoA entering the TAC. The abundance of [U-¹³C]-Palmitate labelled acetyl-CoA decreased in KO mice compared to WT mice. (n = 3 independent heart samples per group).

Figure 5 (i) Quantification results of ¹³C-glucose metabolic flux analysis. F-6-P (m+6), F-1, 6-BP (m+6) and Pyruvate (m+3) were decreased by overexpression KLF7. Results are showed as fractional changes. (n = 3 independent heart samples per group). **(j)** The abundance of [U-¹³C]-Palmitate labelled acetyl-CoA increased in TG mice compared to WT mice. (n = 3 independent heart samples per group).

Representative studies regarding this question are listed here as follows:

[1] Michel, van, Weeghel., et al. Increased cardiac fatty acid oxidation in a mouse model with decreased malonyl-CoA sensitivity of CPT1B. *Cardiovascular research* **114**: 1324-1334 (2018).

Other main points:

1) In response to reviewer 1, the authors point out that other KLFs also induce cardiac

phenotypes - KLF15, KLF10, KLF5 and KLF4. How are they expressed in different cell populations in the heart? How do they change in response to manipulating KLF7?

Author response: Thank you for your valuable suggestion. We highly appreciate your comments. The mouse hearts were digested and sorted into macrophages (CD45⁺F4/80⁺), cardiomyocytes (CD45⁻PDGFR- α ⁻CD31⁻), cardiac fibroblasts (CD45⁻PDGFR- α ⁺), and endothelial cells (CD45⁻ CD31⁺) via fluorescence activated cell sorting (FACS). As shown in the following **Figure 1A and 1B**, the small number of cardiac fibroblasts and macrophages isolated, we quantitatively analyze the KLF4, KLF5, KLF10 and KLF15 mRNA expression in cells was assessed by qRT-PCR in endothelial cells and cardiomyocytes (**Figure 1C**). According to your suggestion, we examined the expression changes of KLFs in KO and TG mouse model and KLF7 knockdown and overexpression NCMs model, respectively. The results are shown in following **Figure 1D-1G**, myocardial knockdown and overexpression of KLF7 resulted in significant changes in the expression of other KLFs, we cannot rule out that KLF7 may interact with other KLFs to play a role in cardiac remodeling, which we will further explore in subsequent experiments. Meanwhile, the changes in the expression of other KLFs may also be induced by KLF7-induced pathologically adverse remodeling in cardiac hypertrophy and heart failure.

Figure 1 (A-B) Flow cytometry sorting of cardiac cell suspension into cardiomyocytes, macrophages, endothelial cells and cardiac fibroblasts, and analysis of the proportion of each cell type. **(C)** KLFs mRNA expression in endothelial cells and cardiomyocytes was assessed by qRT-PCR (n = 6 independent experiments). **(D-G)** KLFs mRNA expression in KO or TG mouse cardiac tissues and NCMs treated with knockdown or overexpression KLF7 was assessed by qRT-PCR (n = 6 independent experiments).

2) A more fundamental question is relating to your rationale for making a cardiac-specific KLF7 KO when the human single cell RNA seq data seem to show that KLF7 is barely expressed in cardiomyocytes, but predominantly found in endothelial cells? There is also no overlap between KLF7, ACADL and PFKL expression? ACADL does

not seem to be detected either?

Author response: Thank you for your critical and helpful suggestion. Since the sequencing database only provides us with a general guide to variation trends to a certain extent, we have done relevant experiments to verify. We quantitatively analyzed the expression of KLF7, PFKL and ACADL in flow cytometry-sorted cardiomyocytes and endothelial cells. As shown in the following **Figure 2**, KLF7, PFKL, and ACADL are expressed in both cardiomyocytes and endothelial cells, among which KLF7 and ACADL are higher expressed in cardiomyocytes.

Figure 2 Flow cytometry sorting of mice cardiac single cell suspension into cardiomyocytes and endothelial cells, the KLF7, PFKL, and ACADL mRNA levels were determined by qRT-PCR (n = 6 independent experiments).

3) "Fig 1 (a) Analysis of the GEO single-cell sequencing database revealed a decrease in KLF7 expression. (b, c) increase in PFKL expression and decrease in ACADL expression in dilated cardiomyopathy patients (n = 11492 cells). "

This change is not apparent from those figures. All I see is a huge variability of the expression levels.

Author response: Thank you for your critical and helpful suggestion. Considering your suggestion, we deleted the data from we chose human single-cell database and used human adult cardiomyocyte cell line AC16 treated with AngII to induce cardiomyocyte hypertrophy for relevant verification. The results showed that compared with the PBS

group, the expression of BNP was significantly increased in AngII group, revealing that the AC16 cardiomyocyte hypertrophy model was successfully constructed. As we expected, cardiomyocyte hypertrophy resulted in a significant decrease in KLF7 and ACADL expression and a significant increase in PFKL expression (**Figure 1a-1c in the revised manuscript**).

The following description has been added in the revised manuscript and marked in red.

To clarify whether KLF7 is involved in the process of cardiac hypertrophy, we used AngII to treat human cardiomyocyte cell line AC16 to induce cardiomyocyte hypertrophy model, and the results showed that the expression of KLF7 decreased in hypertrophic cardiomyocytes (**Figure 1a, 1b**).

This finding was also verified quantitative analysis the PFKL and ACADL mRNA levels in AC16 cell line treated with AngII stimulation (**Figure 1c**).

Figure1 (a-c) Quantitative analysis of the mRNA levels of BNP, KLF7, PFKL and ACADL in human cardiomyocyte cell line AC16 treated with AngII for 24h (n = 6 independent experiments).

4) The Western blots are of very poor quality. I would consider all the blots for ACADL invalid - smearing of band makes these readouts not reliable for quantitation.

Author response: Thank you for your valuable suggestion. According to your suggestion, we re-determined all western blot experiment involving ACADL in our

paper and improved the equality of the protein bands, as shown in **Figure 1h, 1i, Figure 3b, Figure 5c, Supplemental Data 3g, 3h, 10h** in the revised manuscript.

Figure 1 (h) Cardiac hypertrophic growth induced by TAC downregulated the expression of ACADL at the protein level (n = 7 independent experiments). **(i)** The protein level of ACADL was downregulated in NMCs 48 hours after AngII treatment (n = 5 independent experiments).

Figure 3 (b) KLF7 deficiency downregulated ACADL at the mRNA and protein level (n = 4-7 independent experiments).

Figure 5 (c) The protein level of ACADL was upregulated in myocardial tissues of TG mice of different ages compared to their expression in corresponding WT mice (n = 7 independent experiments).

Supplemental Data 3 (g) Western blot analysis of ACADL in NMCMs treated with overexpression PFKL (lenti-PFKL) and controls (lenti-NC) (n = 7 independent experiments). **(h)** Protein expression analysis of ACADL expression in knockdown ACADL (sh-ACADL) and controls (sh-NC) (n = 6-8 independent experiments).

Supplemental Data 10 (h) The infection efficiency upon mouse tail-vein injection of adeno-associated virus (AAV) for ACADL expression in myocardial tissue (n = 7 independent experiments).

5) The Western blot in Fig 2b has a MW shift in GAPDH between lanes?! Looking at the raw data in the folder “Western Blot NMCMs+lenti-PFKL” it is apparent that from “GAPDH(BNP).tif” the first four lanes on the original plot were cut and cropped together with the following 5 lanes so they became the last 4 four lanes instead. The cropping was done so poorly that it resulted in a shift in GAPDH as well as ANP. Moreover, GAPDH has a MW of 36kD, it’s annotated correctly as 37kD in the main figure 2, but in the original plot ““Western Blot NMCMs+lenti-PFKL” “GAPDH(ANP).tif” the band is running between 10-25kD instead? The Western blot membranes should be presented in their entirety in the supplement, instead the blots provided are again cropped.

Author response: Thank you for your critical and helpful suggestion. In view of your suggestion, in our paper Fig 2b, the ANP and GAPDH bands do have the problem you mentioned. Therefore, we re-prepared the relevant samples to examine the effect of overexpression of PFKL on ANP protein expression in NMCMs. The result is shown in

the **Figure 2b in the revised manuscript**, compared with the lenti-NC group, overexpression of PFKL did increase the expression of ANP protein. Moreover, we adjusted the molecular weight of all GAPDH in our paper from 37kDa to 36kDa. The uncropped Western blot membranes images in our paper have been presented in the supplement.

Figure 2 (b) Representative western blot images and quantification of upregulated ANP expression in NMCMs infected with lenti-PFKL and lenti-NC. (n = 5 - 7 independent experiments).

6) Suppl Data 3:"(h) Protein expression analysis of ACADL, KLF7 and PFKL expression in knockdown ACADL (sh-ACADL) and controls (sh-NC) (n = 4-5 independent experiments)"

Panel H -the n-numbers for sh-NC and sh-ACADL in the Western blot presented are n=5, densitometry for KLF7 however only shows quantification for n=4 in controls and sh-ACADL group? For PFKL on the other hand, the n-number is n=8-10? How is that possible? Are you referring biological replicates, technical replicates or to independent experiments?

Author response: Thank you for your valuable suggestion. The experimental replicates involved in our paper are the replicates of independent experiments, as you mentioned in Supplemental Data 3h, the n-number of each group is different when the same group

determines different protein expression, the reason is that we did not determine ACADL, KLF7 and PFKL at the same time point, they were detected in different batches of experiments, and the detection of PFKL protein was done by adding the samples of two batches of experiments together, resulting in a larger n-number.

7) Suppl Data 5: The cardiac specific KLF7 KO reduces KLF7 content in heart by 50%. Again, the Western blot is very poor. Staining for KLF7 in the different cardiac cell types would be important. Confirmation of successful KO in cardiomyocyte could be performed in isolated cardiomyocytes with PCR and compared to the non-myocyte fraction.

Author response: Thank you for your valuable suggestion. We performed immunofluorescence co-localization of KLF7 in cardiomyocytes (α -actin), cardiac fibroblasts (Vimentin), endothelial cells (CD31) and macrophages (CD68). The results showed that KLF7 was expressed in the above cells (**Supplemental Data 5l in the revised manuscript**), resulting the knockout efficiency of KLF7 in the hearts of KO mice that we tested was only 50%. Therefore, as you suggested, we used flow cytometry to sort the cardiac cell suspensions of WT and KO mice into macrophages (CD45⁺F4/80⁺), cardiomyocytes (CD45⁻PDGFR- α ⁻CD31⁻), cardiac fibroblasts (CD45⁻PDGFR- α ⁺), and endothelial cells (CD45⁻ CD31⁺) (**Figure 3A-3D**). The number of selected cardiac fibroblasts and macrophages was small, so we verified the knockout efficiency of KLF7 in the sorted endothelial cells and cardiomyocytes. As shown in the **Supplemental Data 5h in the revised manuscript**, the knockout efficiency of KLF7 in cardiomyocytes was about 93%.

The following description has been added in the revised manuscript and marked in red.

We determined KLF7 could co-location in cardiomyocytes, cardiac fibroblasts,

endothelial cells, and macrophages (**Fig. 5l in the Supplemental Data**).

Then, we determined KLF7 knockout efficiency in KO mice in flow cytometry sorted cardiomyocytes, cardiac KLF7 mRNA levels were reduced by approximately 93% in the KLF7^{KO} mice (**Fig. 5h in the Supplemental Data**).

Figure3 (A-D) Flow cytometry sorting of WT and KO mice cardiac cell suspension

into cardiomyocytes, macrophages, endothelial cells and cardiac fibroblasts, and analysis of the proportion of each cell type.

Supplemental Data 5 (h) KLF7 knockout efficiency in KO mice was about 93% in flow cytometry sorted cardiomyocytes was assessed by qRT-PCR (n = 6 independent experiments). **(I)** Hearts of WT mice were immunofluorescent stained to determine of KLF7 (red) in α -actinin (cardiomyocytes), Vimentin (cardiac fibroblasts), CD31 (endothelial cells) and CD68 (macrophages)-positive cells. Scale bar = 10 μ m.

8) Suppl Data 6 :

KLF7 knock-down in NMCs. How high is KLF7 expressed in NMCs? How does it compare to adult myocytes? How much is it reduced with sh-KLF7? As mentioned above, what are the effects of manipulating KLF7 on other KLF family members?

Author response: Thank you for your generous suggestion. We quantitatively analyzed the expression of KLF7 in flow cytometry sorted cardiomyocytes and NMCs, and the result showed that KLF7 was higher expressed in NMCs. As shown in **Figure 1F**, the knockdown efficiency of KLF7 in NMCs was about 60%.

Then, we examined the expression changes of other KLFs in KO and TG mouse model

and KLF7 knockdown and overexpression NMCMS model, respectively. The results are shown in **Figure 1D-1G**, myocardial knockdown and overexpression of KLF7 resulted in significant changes in the expression of other KLFs, we cannot rule out that KLF7 may interact with other KLFs to play a role in cardiac remodeling, which we will further explore in subsequent experiments. Meanwhile, the changes in the expression of other KLFs may also be induced by KLF7-induced pathologically adverse remodeling in cardiac hypertrophy and heart failure.

Figure 4 Quantitative PCR analysis of KLF7 mRNA expression in flow cytometry-sorted cardiomyocytes and NMCMS (n = 6 independent experiments).

Figure1 (D-G) KLFs mRNA expression in KO or TG mouse cardiac tissues and NMCMS treated with knockdown or overexpression KLF7 was assessed by qRT-PCR (n = 6 independent experiments).

REVIEWER COMMENTS

Reviewer #3 (Remarks to the Author):

The authors have adequately addressed my remaining concerns. However, I recommend the remaining minor change:

The sentence "The empaglifloxin reduce glycolysis, rebalance coupling between glycolysis and oxidative phosphorylation to attenuate adverse cardiac remodeling and progression of HF induced by pressure-overload⁶²." should be changed to "Empagliflozin reduces glycolysis and rebalances coupling between glycolysis and oxidative phosphorylation to attenuate adverse cardiac remodeling and progression of HF induced by pressure-overload⁶²."

Reviewer #4 (Remarks to the Author):

The authors have made important additions that improved the manuscript.

My only minor comment relates to this statement in the method section:

"Samples containing an equal amount of protein (10-220 µg) were separated on a 12% SDS-PAGE gel (Thermo), transferred to a nitrocellulose membrane, and immunoblotted, after which specific protein bands were detected and quantified using an Odyssey scanner (LI-COR version 3)."

I suppose 220 µg is a typo? This would overload the Western blot. Also, it would be advisable to scan the entire membrane not just the height of the bands of interest.

Reply to Reviewer #3

Comments from Reviewer #3: The authors have adequately addressed my remaining concerns. However, I recommend the remaining minor change:

The sentence "The empagliflozin reduce glycolysis, rebalance coupling between glycolysis and oxidative phosphorylation to attenuate adverse cardiac remodeling and progression of HF induced by pressure-overload⁶²." should be changed to "Empagliflozin reduces glycolysis and rebalances coupling between glycolysis and oxidative phosphorylation to attenuate adverse cardiac remodeling and progression of HF induced by pressure-overload⁶²."

Author response: Thank you for your valuable suggestion. We have corrected the sentence as you suggested and marked in red in the revised manuscript.

Reply to Reviewer #4

Comments from Reviewer #4: The authors have made important additions that improved the manuscript.

My only minor comment relates to this statement in the method section:

"Samples containing an equal amount of protein (10-220 µg) were separated on a 12% SDS-PAGE gel (Thermo), transferred to a nitrocellulose membrane, and immunoblotted, after which specific protein bands were detected and quantified using an Odyssey scanner (LI-COR version 3)."

I suppose 220 µg is a typo? This would overload the Western blot. Also, it would be advisable to scan the entire membrane not just the height of the bands of interest.

Author response: Thank you for your critical and helpful suggestion. Sorry, it's indeed a typo error, we have corrected as 10-20 µg and marked red in the *Method* section of

the revised manuscript. Furthermore, we repeated and quantified the western blot experiments involved in our paper and minimized the cut membrane as much as possible. All the original images are in the “Source Data” folder and named as “Western Blot”. At the same time, we also tried to scan the entire membrane to incubate the antibody of the target protein and the reference protein reduce the cutting, but the different exposure times of different antibodies would lead to the phenomenon as shown in the figure below. The blot was unclear and the band was poor.